# Competition, capital growth and risk-taking in emerging markets: Policy implications for banking sector stability during COVID-19 pandemic

**Miroslav Mateev**[1]*, **Muhammad Usman Tariq**[1], **Ahmad Sahyouni**[2]

**1** Abu Dhabi School of Management, Abu Dhabi, UAE, **2** Higher Institute for Administrative Development, Damascus University, Damascus, Syria

☉ These authors contributed equally to this work.
* m.mateev@adsm.ac.ae

**Data Availability Statement:** All relevant data are within the manuscript and its Supporting Information files.

## Abstract

This paper investigates how banking competition and capital level impact on the risk-taking behavior of banking institutions in the Middle East and North Africa (MENA) region. The topic is perceived to be of significant importance during the COVID-19 pandemic. We use data for more than 225 banks in 18 countries in the MENA region to test whether increased competition causes banks to hold higher capital ratios. Employing panel data techniques, and distinguishing between Islamic and conventional banks, we show that banks tend to hold higher capital ratios when operating in a more competitive environment. We also provide evidence that banks in the MENA region increase their capitalization levels in response to a higher risk and vice versa. Further, banking concentration (measured by the HH-index) and credit risk have a significant and positive impact on capital ratios of IBs, whereas competition does play a restrictive role in determining the level of their capital. The results hold when controlling for ownership structure, regulatory and institutional environment, bank-specific and macroeconomic characteristics. Our findings inform regulatory authorities concerned with improving the financial stability of banking sector in the MENA region to strengthen their policies in order to force banks to better align with capital requirements and risk during the COVID-19 pandemic.

## Introduction

In this study we investigate the impact of banking competition and the level of risk-taking on capital ratios of banks in the countries that belong to the Middle East and North Africa (MENA) region. The excess risk-taking by banking institutions is considered by many researchers as the key factor contributing to the financial crisis of 2007–2008, which forced some countries to adopt strategies to increase the level of concentration and reduce the banking sector competition in order to increase financial stability [1]. In this context, the importance of market competition as an explanatory factor for banking soundness, and, more specifically, bank capital ratios, has significantly increased. However, both economic theory

**Funding:** The author(s) received no specific funding for this work.

and empirical studies provide contradictory predictions about the relationship between market competition and financial stability in the banking system.

The relationship between banking competition and financial stability has been an important topic with much debates in the economic literature [2]. The main stream in the literature supports the competition-fragility hypothesis. Under this hypothesis, banking competition will lower the interest income for banks and therefore, banks' profits will decrease, which will lead to increased probability of default, and consequently, an overall disruption of the financial system [3–5]. On the other hand, supporters of competition-stability hypothesis [6–8], think that banks with more market power tend to increase their interest rates; in turn, high rates of interest can lead to moral hazard problem by increasing non-performing loan ratio of banks. So under this theory, competition increases the financial stability.

Empirical literature that investigates the effect of competition and concentration on banking stability also provides conflicting evidence. For example, earlier work implies an inverse relationship between competition and banking stability [9]. However, a growing body of empirical research suggests that increased competition, increased concentration, and sectors with greater contestability and less activity restrictions, are all associated with banking stability [7, 8, 10–13]. This analysis is even more relevant for emerging economies. For example [14], study the extent of bank competition in the MENA region during 1994–2008, using the H-Statistic and the Lerner index. Their analysis suggests that banking sector competition in the MENA region is lower relative to other regions and has not improved in recent years. Furthermore, the authors argue that the lower levels of competition in the region are explained by the region's worse credit information environment and lower market contestability. More recently [2], examines and compares the behavior of different banking systems (Islamic and conventional) in the MENA region in relation to the impact of capital adequacy ratio on bank risk-taking in different competitive circumstances. The study reports that the capital ratio has a significant impact on the behavior of both types of banking systems, whereas, the competitive conditions have no effect on the relationship between the risk-weighted asset ratio and Islamic banks' behavior.

The role of capital requirements for the financial stability of banking systems during the COVID-19 pandemic become of significant importance for researchers. Earlier research [15] claims that regulatory capital requirements act as a safeguard of risk and improve the performance and efficiency of banks. Similarly [16], find that capital requirements have a significant impact on lending activities of banks and consider capital as a shock absorber of credit risk. Other similar studies that report a negative relationship between risk and capital also indicate capital as an effective tool for managing risk (see e.g., [17–24], among others). Thus, the regulatory pressure of implementing Basel III capital regulations and the existing literature both support the notion that capital regulation tends to improve bank efficiency and enhances bank protection against risk. The recent studies on COVID-19 pandemic investigate predominantly the government interventions on stock market return [25–27]. The studies that examine the impact of the pandemic on bank performance (more specifically, the financial stability of banks) support the notion that during the financial crisis the responses of capital to risk-taking aptitude of banks is not similar to the normal economic conditions [27, 28]. For example [29], finds that higher capital levels promote banks' financial stability by lessening the risk, and higher risk impedes the growth of capital. He concludes that during the new global crisis (COVID-19) bank requires more capital to absorb shocks, and COVID-19 pandemic also hits hard banks' capacity of survival.

Our analysis of the existing empirical literature indicates that there is no unambiguous answer to the question of whether banks raise their capitalization levels in response to a higher risk or vice versa [30]. This literature claims that competition and risk-taking may affect the

level of capital requirements but capital, in turn, may induce the bank to take on more risk. For example [31], find that competition decreases the probability of default of a single loan, while capital regulation increases it. The reason is that stricter regulation decreases the competition for loans, implying higher interest rates, and hence greater risk-taking incentives to borrowers. Similarly, more stringent capital requirements may induce the bank to choose a higher level of credit risk. In this case, more rigorous capital requirements can lead to a higher probability of default of banks. Market conditions that yield a risk-decreasing effect of competition tend to cause a risk-increasing effect of capital regulation. Therefore, if there is a trade-off between competition and financial stability, then capital tends to have a positive effect on bank stability. If, however, limiting the number of bank charters weakens the banking system, then capital regulations tend to have inverse effect on stability; in other words, stricter capital requirements will tend to increase the probability of bank failure [32].

In this paper, we empirically examine the impact of risk and market competition on capital ratios using bank-level data. More specifically, we analyze the relation between market competition (using two competition measures, the H-Statistic and the HH-Index) and capital ratios for more than 225 banks in 18 MENA counties, over the period of 2006–2018.

Our study differs from previous research on the MENA region in two ways. First, while most of the previous studies investigate the trade-off between banking competition, capital and risk-taking mostly in global or Europe-specific aspect, we focus on emerging markets in the MENA region. The reason is that [14], among others, find that MENA banking sectors operate under monopolistic competition and Gulf Cooperation Council's (GCC) countries tend to be less competitive than non-oil producing countries. Employing panel data techniques, and distinguishing between Islamic and conventional banking systems, we demonstrate that banks tend to hold higher capital ratios when operating in a more competitive environment. Furthermore, we show that banks in the MENA region raise their capitalization levels in response to a higher risk. Second, previous research does not distinguish this effect between Conventional banks (CBs) and Islamic banks (IBs). For example [2], find that capital ratio has a significant impact on the credit risk behavior of both CBs and IBs, whereas the competitive conditions have no effect on the relationship between the risk-weighted asset ratio and IBs' credit risk. However, this and other similar studies that incorporate capital requirements effect, do not examine the risk (financial stability) impact on banks' capitalization level. Using a sample of 162 CBs and 63 IBs in 18 MENA countries, this study finds that the impact of risk and market competition on capital ratios is significantly different between Islamic and conventional banking systems.

We contribute to the existing empirical literature in several ways. First, the issue of increased role of banking competition and risk-taking for the level of bank capital is not well addressed in previous studies. For example [1], examines the relationship between market competition and bank stability of 356 banks operating in the MENA region, and finds that in less-competitive markets, increased competition may favor the risk-shifting effect and help improve efficiency, which in turn improves financial stability. Our study takes a different approach. Using data for more than 225 banks in 18 countries in the MENA region, we test the hypothesis that both risk and increased competition causes banks to hold higher capital ratios. Our analysis shows that banks increase their capitalization level in highly concentrated markets, whereas, market competition (measured by the H-Statistic) has a limited effect on capital ratios of banks. The increased risk also has a positive impact on the level of bank capital.

Second, previous research (see e.g., [30] that explores the relationship between capital and risk reached to the conclusion that banks raise their capitalization levels in response to a higher risk rather than the other way round. These studies report that, on average, IBs are more stable

(and therefore, less risky) as compared to CBs. However, the effect of banking competition on IBs behavior remains unexplored. We complement this research by investigating the role of banking competition and concentration separately for CBs and IBs. Our analysis finds that banking concentration (measured by the HH-index) has a significant positive impact on capital ratios of either bank, whereas increased competition does play a restrictive role for IBs on increasing the level of capital. Our findings inform the regulatory authorities concerned with improving the financial stability of banking sector in the MENA region during the COVID-19 pandemic for the need to strengthen their policies that force banks to better align with capital requirements and risk in considering the level of market competition.

Third, previous research finds that more rigorous capital requirements can lead to a higher probability of default of banks. For example [31, 32], find that competition decreases the probability of default of a bank loan, but capital regulation increases it. We provide new evidence for emerging markets in the MENA region. More specifically, for the conventional banking sector, competition has a negative effect on bank credit risk, while in the group of Islamic banking institutions, this effect is insignificant. However, banking concentration (measured by the HH-index) does impact on IBs' risk behavior. Our results also indicate that banks in the MENA region increase their credit risk in response to an increase in capitalization level to meet regulatory requirements. These findings are important for regulators and policy makers which can set capital requirements at level that would restrain banks from taking excessive risk, depending on the level of ownership and banking sector concentration.

The contribution of our paper is also related to the growing empirical literature that studies the economic impact of the pandemic on bank performance. For example [33], analyze bank stock prices around the world (including the MENA region) to assess the impact of the COVID-19 pandemic on the banking sector. Using a global database of policy responses during the crisis, the paper also examines the role of financial sector policy announcements on the performance of bank stocks. The results show that the impact of prudential measures (which deal with the temporary relaxation of regulatory and supervisory requirements, including capital buffers) appeared to be limited, except in countries that are not part of the Basel Committee, where such policy initiatives have a negative impact on bank returns. In line with this finding, our paper suggests that regulators responsible for banking sector stability should require a more disciplined approach in bank lending decisions and building sufficient capital conservation' buffer to limit the impact of downside risk from depletion of capital buffers which is perceived to be significant during the pandemic.

Our paper is organized as follows: Section 2 presents the literature review and formulates the hypotheses. Section 3 describes the data set and the research methodology. Section 4 contains the results of the empirical analysis and the interpretations. Section 5 includes a robustness check and alternative specifications. Finally, Section 6 details our conclusions.

## Theoretical framework and hypotheses

### Market competition and bank stability theories

There are two opposing theories regarding the impact of market competition on bank behavior [2]. The first theory argues for a positive relation, that is, a competitive market may increase banks' risk-taking behavior in order to maintain their previous level of profit [3]. This risky behavior can be noticed either through the increase in credit risk of the loan portfolio, or through the fall in the level of capital buffer, or simultaneously. Such behavior can lead to an increased level of non-performing loans and subsequently a greater probability of bank default. The second theory postulates that a restricted competition should encourage banks to protect their high "franchise value" by pursuing safety strategies that contribute to the stability of the

whole banking system [2]. Therefore, according to the paradigm of the franchise value, banks limit their risk when they have market power in lending (however, according to [34] the traditional view that high franchise value reduces bank risk-taking incentives does not always hold). A study by [35] provides further support to the 'franchise value' paradigm in limiting bank risk-taking. As the underlying source of franchise value is assumed to be the market power of a bank, reduced competition among banks has been considered important to promote banking stability. Conversely, an increase in banking competition erodes their quasi-monopoly rents. The results of the study using Lerner index based on bank-specific interest rates, indicate a negative relationship between market power and bank risk-taking, which lends further supports in favor of the 'franchise value' theory.

Empirical literature also provides conflicting evidence on the implications of increased competition for bank stability. For example [4, 36], suggest that increased competition decreases banks' soundness. The key assumption in this notion is that bank managers have an incentive to take excessive risk so as to benefit shareholders at the expense of depositors. In opposite [37], demonstrate that monopoly banks with intermediate monitoring costs can be more prone to rent risky loans that give rise to a higher probability of default. In the same context [6, 7], suggest that allowing for competition in lending markets is likely to increase bank stability, whereas [3] highlight that the relationship between competition and stability in the banking sector is multilayered, with no simple trade-off between the two. Most of the previous research arrived to the conclusion that "more competition is ceteris paribus associated with a lower probability of failure. In other words, there is a positive relationship between competition and bank stability" [1].

Similar results are reported by [38] who examine the impact of banking competition on systemic stability, using a sample of 1,872 publicly traded banks in 63 countries. Their evidence supports the so-called competition-stability viewpoint. The competition-stability theory or risk-shifting view (see [6–8]) suggests that banks with more market power tend to charge higher interest rates, which provides an incentive to borrowers to engage in risky activities, which makes it more likely that the borrower will default on its obligation. So, under this theory, more competition increases the financial stability. In line with this [38], argue that increased competition will induce banks to take more diversified risks and therefore, the banking system will be more resilient to shocks. [13] show that the contradictory evidence reported by previous empirical studies is attributable to the way competition has been measured. These studies have been usually based on the 'structure-conduct-performance' paradigm, which assumes that market structure is related to competitive conduct and that competition can be approximated by the degree of concentration in the banking sector. The measures of concentration are usually computed using country-level concentration ratios [11, 12].

However, according to the industrial organization literature, measures of market structure such as the number of institutions and concentration ratios, are not necessarily related to the level of competitiveness in an industry [39]. For the same reason [40], conclude that it is inappropriate to rely on concentration to assess the degree of competition in banking sector and that more research is needed. However, the so-called 'efficient-structure' theory, assumes that more efficient firms tend to operate at lower costs and therefore increase market share [41]. This hypothesis is based on the premise that firms with low cost structures increase their profits by reducing prices and expanding market shares. Therefore, a positive relationship between firm profits and market structure exists because of gains made in market share by more efficient firms. In turn, these gains lead to increased market concentration

The existing evidences from the emerging markets in the MENA region are scarce. Most of the existing research [1, 2] investigate the trade-off between market structure and risk-taking behavior of banks using either structural or non-structural approach to measuring bank

competition. The relationship between market competition, financial risk and capitalization level of banks in the MENA region remains unexplored. Another aspect of banks' risk attitude and consequently, the level of asymmetric information of their loan and/or security portfolio, is the relationship banking.

The research on relationship lending in the MENA region is very limited and provides inconclusive evidence. For example, a study by [42] reports that banks in the MENA region still seem to rely on relationship lending, possibly to compensate for the weak financial infrastructure and information asymmetries. The analysis of distribution channels used by banks to service SMEs points to the importance of branches offering services that are tailored to SME needs, which may reflect the continuing importance of 'relationship banking'. However, it is not clear if the presence of an SME unit by itself means that the bank has moved from relationship lending to transactional lending. The study concludes that banks use most probably the relationship lending to overcome information asymmetries and the opaqueness of SMEs in the MENA region. In the same context [43], investigate whether borrowers enjoy the bright side or suffer the dark side of their banking relationships during the COVID-19 crisis. Their results are consistent with the empirical dominance of the dark side of relationships during the crisis; these findings hold across different loan contract terms, relationship measures, COVID-19 shocks, and loan types. The conclusion is that banks do not appear to be "friends indeed with their relationship borrowers in need" [43].

## The trade-off between competition, level of bank capital and risk-taking

Banks are seen as the most important financial institutions that provide markets with liquidity [44]. The optimal level of the allocated capital should take into account the mandatory control imposed by the regulators since the banking sector is one of the most regulated industries in the world. Bank regulation is primarily based on the minimum capital requirements set by the Basel Committee to strengthen the stability of the banking system and reduce bank risk. Therefore, all banks today are subject to minimum regulatory capital requirements set up by Basel II guidelines [45].

However, there are important reasons for banks to hold more than the required minimum. Theoretical studies suggest that competition may be one of the reasons for banks doing so. For example [46], develop a model in which commercial banks compete through setting acceptance criteria for granting loans. By making easier such criteria, a bank faces the trade-off between attracting a greater demand for loans, thus making higher profits, and deteriorating the quality of its loan portfolio, therefore, bearing a higher risk of default. One of the results of this model is that it is beneficial for a bank to hold more equity in a competitive environment than prescribed by the regulator, even though issuing equity is more expensive than attracting deposits. In the same context [47], build a model suggesting that equity capital may be higher in situations with highly competitive credit markets when good lending opportunities are scarce. Identifying the relationship between bank capital and risk [48], argue that capital over minimum requirements induces banks to take more risk that results in high amount of non-performing loans. As equity capital is a costly source of financing [4], maintaining more capital over the minimum capital requirements may affect the performance of banks in terms of profitability and efficiency.

The empirical literature also investigates the association between market competition, level of capital, and risk. For example [49], opine that with the decrease of market competition banks will hold higher capital ratios but their risk-taking will also increase and that tends to increase the probability of default. Studies on different regions have reached similar results. For example [50], report that cooperative banks in Europe show a tendency to increase their

risk-taking in less competitive markets. [51] study the Asia Pacific region and find that more concentration enhances bank fragility and risk measured by distance-to-default (or Z-score) of banks. In other words, the reduction of market competition increases the probability of bank default. However, a growing body of empirical evidence supports the tendency of taking more risk with the increase of market competition in a banking sector (see for example [4, 36], among others). Competition not only increases risk-taking but may also affect the level of capital requirements; in turn, capital may induce the bank to take on more risk. In support of this notion [31], report that market conditions that yield a risk-decreasing effect of competition tend to cause a risk-increasing effect of capital regulation. A more recent study by [52], using data from 167 banks in 37 African countries, reaches the conclusion that regulatory capital plays no significant role in enhancing financial stability and overall competition of banks. The authors also claim that increased regulatory requirement give competitive advantages to foreign banks because of the low cost of capital sourcing. However, due to the high cost of capital, domestic banks become less competitive.

This analysis of emerging economies in the MENA region also finds conflicting evidence. For example [53], find no significant relationship between capital stringency and the likelihood of bank distress in the GCC region. In contrast [54], finds that the implementation of the Basel II capital regulation has a positive effect on credit growth of banks in Egypt, Jordan, Lebanon, Morocco and Tunisia. [30] explores the relationship between capital and risk in 57 CBs and 46 IBs in the MENA region. The results indicate that banks raise their capitalization levels in response to a higher risk rather than the other way round. Using a sample of 52 IBs and 186 CBs in 14 Organization of Islamic Conference (OIC) countries from 1999 to 2009 [55], find that there is a significant and positive relationship between capital adequacy ratio and banking activity. A more recent study of [1] empirically examine the relationship between market competition and risk-taking behavior of banks in the MENA region. They find that, in countries where the level of competition is high (e.g., Gulf countries), the rise in competition increases the probability of default; however, when the level of competition is low (e.g., in non-Gulf countries), the increase in rivalry can be positive in terms of risk-shifting and efficiency. However, their study does not differentiate between Islamic and Conventional banking systems. Previous research reports that IBs are more stable (and therefore, less risky) as compared to CBs as they hold higher level of capital. However, the impact of market competition on bank capital and the risk behavior of IBs and CBs in the MENA region remains unclear. To shed more light on this issue, we develop and test the following hypotheses:

*Hypothesis 1a*: *Market competition and risk have a significant impact on bank capital ratios.*

*Hypothesis 1b*: *Market competition and capital ratios have a significant impact on bank risk.*

## Is there a differential impact of market competition on IBs risk behavior?

While the majority of previous studies explain the difference in Islamic and conventional bank behaviors with the fact that IBs operate in accordance with the principles of Sharia, others confirmed that IBs diverge from their theoretical models by adopting CBs' strategies. In this context [56], argues that IBs' activities are based on sales instruments rather than on partnership. [57] point at the fact that Islamic financial institutions face extra risk because they have limitations in financing, investing and risk management activities, and, at the same time, their financial practices are more complex. Another strand of the literature (see e.g., [58]) stipulate that, since both types of banks operate in the same competitive environment and are regulated in the same way in most countries, they are likely to have a similar behavior and thus similar risk strategies. [59] also argue that in practice, IBs are not different than CBs, and suggests that the

fast growth of Islamic banking is not due to the principals of Sharia-compliant banking but rather to Islamic resurgence worldwide.

Islamic products tend to be more complicated than their conventional counterparts since they usually involve more than one concept and non-standard transaction structures. Taking into the account the nature of financial contracts used by Islamic banks or conventional banks with Islamic windows, such as *musharaka*, one may expect that the information used in the screening process of clients to be largely soft, which raises both the risk and costs incurred by the bank when assessing their clients. Relationship banking is one possible approach to resolve this problem as it enables better monitoring and screening of borrower [60]. Besides, banks can use their comparative advantage when monitoring clients. Small size banks tend to have the benefit of accessing and processing soft information about small and medium enterprises (SMEs), while large banks are more skillful in screening large enterprises because of their economies of scale and scope. Taking into account that the majority of Islamic banks are small, it is likely that their information advantage is mainly with SMEs. However, SMEs tend to produce and reveal less information compared to large firms and hence, Islamic banks need to be more cautious when lending to SME to overcome the issue with asymmetric information.

In Islamic financial contracts the level of asymmetric information tends to vary from one contract to another. [61] states that one of the main reasons partnership (*musharaka*)-based contracts are less attractive is the high asymmetric information attached to these contracts compared to other financial agreements. High banking reserves and capital provide more assurance to banks against asymmetric information, particularly for those who are placing their funds under less restricted financial contracts like *mudaraba*. Moreover [60], suggests that Islamic financial contracts are subject to different type of asymmetric information (*gharar*) -related problems at both the ex-ante and ex-post stages of the lending process. Therefore, taking into account the portfolio of Islamic banks and in order to minimize losses caused by the asymmetric information, Islamic banks need to use more secure financing, particularly with SMEs, which tend to be more financially vulnerable when the economy is in decline. Islamic banks can also adjust their loan pricing to reflect the new lending risks (e.g., credit and market risk) and pass some of their costs to borrowers.

Previous empirical research does not provide explicit answer to the question of whether there is a differential impact of market competition on IBs behavior. For example [62], examine a sample of banks operating in 17 countries in the MENA region where IBs and CBs coexist. The study measures and compares the market power of the Islamic and conventional banks by calculating the Lerner index, and finds no significant difference between IBs and CBs, over the period of 2000–2007. However, regressions including control variables indicate that IBs have less market power than CBs. Thus, any reduced market power of IBs can be attributed to differences in the business model and the risk management practices employed by these banks. [2] find that competitive conditions have no significant effect on the relationship between capital adequacy ratio and IBs' risk behavior, which means that this type of banks are still applying theoretical models based on the prohibition of interest. More recently [1], argue that in markets with a high degree of competition, increasing this level further may have an effect on the margin of interest that does not offset the risk-sifting effect, and suggest a U-shaped relationship between competition and risk of failure for MENA banks. In fact, banks operating in the MENA countries have a moderate level of competition, so we may expect that the relationship between competition and risk-taking behavior can be explained by the competition-fragility hypothesis.

Another group of studies investigate the impact of market competition on capital ratios of banks in emerging markets. For example [49], use a sample of 636 commercial banks in 11

Asian countries to explore the impact of market power on bank capital ratios, income volatility, and insolvency risk. The analysis indicates that higher degree of market power is associated with higher capital ratios, higher risk-taking and increased insolvency risk. However, this effect does not hold during the 1997 Asian crisis where higher market power is associated with less risk-taking by banks, and therefore better financial stability. A study on MENA region by [2] investigates 70 CBs and 47 IBs in 12 MENA countries, and finds that competitive conditions have no significant effect on the relationship between capital adequacy ratio and IBs' risk behavior, which means that this type of banks are still applying theoretical models based on the prohibition of interest. Moreover, it turned out that the behavior of Islamic banks is independent from the level of market competitiveness and therefore, from the interest rate. Hence, the study concludes that both banking sectors have different behavior showing that Islamic banks are still applying their theoretical models.

We complement these findings by exploring the impact of increased competition, increased concentration and risk-taking on capital ratios of banks in the MENA region. Based on the fact that Islamic banking market is more concentrated [63], we expect a strong differential impact on IBs behavior to exist. More specifically, we formulate and test the following two hypotheses:

*Hypothesis 2a. The impact of market concentration on bank capital and risk is expected to be different between Islamic and conventional banks.*

*Hypothesis 2b. The impact of market competition on bank capital and risk is expected to be similar between Islamic and conventional banks*

## Data and methods

### Sample selection

We use a data set that covers 2,489 observations from 225 banks in 18 MENA countries, including the six GCC countries (Bahrain, Kuwait, Oman, Qatar, Saudi Arabia and United Arab Emirates), over a period of 14 years (2005–2018). The accounting data are collected from the database of Orbis Bank Focus (Bureau Van Dijk), together with the annual reports of the banks included in the sample. The period of analysis represents the years for which accounting data are currently available for all banks in our sample either Islamic or conventional. Moreover, we use other sources of secondary data such as the World Bank's Worldwide Governance Indicators (WGI) [64] and World Development Indicators (WDI) [65], International Financial Statistics and annual reports of the central banks to collect macroeconomic data. The selection of the sample period to cover the years from 2005 to 2018 is dictated by the data availability for banks in the sample for each year of the observation period. The data before 2005 is incomplete or even missing for some banks, so it has been excluded from the analysis. A detailed description of the dependent and independent variables is provided in S1 Table in S1 Appendix.

Our sample contains both Islamic and conventional banking institutions in 18 countries in the MENA region. Table 1 provides sample statistics that includes the total number of observations for each country, and the number of observations for the sample of IBs and CBs, respectively (**see S1 Data**). The data indicate that 162 banks (or 72.0%) in the sample are conventional banks and the rest are Islamic banks. In addition, Table 1 contains information for H-Statistic for IBs and CBs, across the countries. The H-Statistics indicate that the banking systems in the sample are characterized by monopolistic competition. While Iran, Iraq, Syrian Arab Republic and Palestine exhibit comparatively low levels of competition, Bahrain, Lebanon and Unite Arab Emirates appear to have the most competitive banking systems in the MENA region. We winsorize the bank-level explanatory variables at the 1% and 99% levels.

**Table 1. Types of banks and H-Statistic per country.**

| Country | Observations (All banks) | Observations (Conventional) | Observations (Islamic) | Number of Conventional Banks | Number of Islamic Banks | H-Statistic country-wide) |
|---|---|---|---|---|---|---|
| Algeria | 105 | 105 | 0 | 9 | 0 | 0.64 |
| Bahrain | 231 | 112 | 119 | 9 | 11 | 0.87 |
| Egypt | 303 | 277 | 26 | 23 | 2 | 0.49 |
| Iran (Islamic Republic of) | 98 | 0 | 98 | 0 | 11 | 0.32 |
| Iraq | 71 | 53 | 18 | 5 | 2 | 0.13 |
| Israel | 98 | 98 | 0 | 8 | 0 | 0.32 |
| Jordan | 181 | 150 | 31 | 12 | 3 | 0.35 |
| Kuwait | 127 | 64 | 63 | 5 | 6 | 0.36 |
| Lebanon | 241 | 237 | 4 | 22 | 1 | 0.65 |
| Morocco | 93 | 93 | 0 | 8 | 0 | 0.47 |
| Oman | 82 | 75 | 7 | 6 | 1 | 0.38 |
| Palestinian Territory | 41 | 18 | 23 | 2 | 2 | 0.13 |
| Qatar | 121 | 76 | 45 | 6 | 4 | 0.38 |
| Saudi Arabia | 132 | 101 | 31 | 8 | 5 | 0.53 |
| Syrian Arab Republic | 121 | 92 | 29 | 9 | 3 | 0.09 |
| Tunisia | 154 | 139 | 15 | 11 | 2 | 0.44 |
| United Arab Emirates | 263 | 164 | 99 | 16 | 9 | 0.57 |
| Yemen | 27 | 26 | 1 | 3 | 1 | 0.03 |
| **Total** | **2489** | **1880** | **609** | **162** | **63** | |

The table shows the total number of observations for the whole sample, the number of conventional banks (CBs) and Islamic banks (IBs) per county, the number of observations for each group of banks, and the H-Statistic for CBs and IBs, respectively. The H-Statistic is calculated with the total revenue as dependent variable. The Panzar-Rosse's H-Statistic is designed to discriminate between competitive, monopolistically competitive, and monopolistic markets.

## Empirical specification

A substantial body of literature has examined the variables that determine capital ratios and risk level of banks. Therefore, we include variables that are known to be significant determinants of bank capital and risk, and are expected to differ between CBs and IBs. In this study, we use an unbalanced dynamic panel model and employ the bank-level and country-based characteristics listed in **S1 Appendix** as control variables. We estimate the following empirical model:

$$\gamma it = \beta 0 + \beta 1 \times \gamma i, t-1 + \beta 2 \times Riskit + \beta 3 \times MarkerCompetit + \beta 4 \times Xit + \beta 5 \times Dt + uit \tag{1}$$

$$\gamma\prime it = \beta 0 + \beta 1 \times \gamma i, t-1 + \beta 2 \times Capitalit + \beta 3 \times MarkerCompetit + \beta 4 \times Xit + \beta 5 \times Dt + uit \tag{2}$$

In model (1), $\gamma_{it}$ is the capital ratio of bank $i$ in year $t$, $Risk_{it}$ and $MarketCompet_{it}$ are the explanatory variables (respectively, bank credit risk and market competition indicator), $X_{it}$ is the vector of control variables (bank accounting ratios and macroeconomic indicators), $\beta_1$ to $\beta_5$ are the regression coefficients, and $u_{it}$ is the disturbance term that is assumed to be normally distributed with a mean of zero. The vector of dummy variables ($D_t$) includes the Islamic

dummy (ISLAMIC) that equals one if a bank is Islamic banking institution and 0 otherwise, and the Crisis time dummy (crisis) that takes the value of one for the years 2008–2009, and 0 otherwise. We follow the work of [66, 67] consider 2008–2009 as the crisis period for the MENA region. In addition to Eq (1), we examine the effect of capital ratios and market competition on risk-taking behavior of banks in our sample. Therefore, Eq (2) incorporates different measures of a bank's capitalization level (the ratio of total eligible capital to total assets, EC/TA and the ratio of total equity to total assets, TE/TA), and the respective market competition indicators (HH-index and H-Statistic), as well as all the explanatory and control variables of Eq (1).

We use fixed effect/random effect specifications and perform a Hausman test where the null hypothesis is that the preferred model is random effects vs. the alternative fixed effects. The choice between random and fixed effects specification depend on the Prob>chi^2 being more or less than 5%, respectively. We also estimate the level of correlation amongst capital ratio, market competition indicator and other important variables to identify if there is any multicollinearity problem. No significant correlation between capitalization variable and market competition measures, and between capital ratio/competition indicators and other important variables is observed. Therefore, the correlation matrix (available on request) suggests that our estimation results do not seem to suffer from multicollinearity problem.

### Dependent variables

In this study, we examine credit risk effect and capital ratios of banks in the MENA region. We measure credit risk using the ratio of Loan loss reserves to gross loans (LLR/GL). This ratio measures loan quality [66, 68, 69], with higher values indicating poorer loan quality or higher protection against credit default risk. For robustness purposes, we also apply Non-performing loans to gross loans (NPL/GL) ratio [1]. Regarding capital ratios, we follow [32] and use the ratio of total eligible capital to total assets (EC/TA) as a proxy for a bank's capitalization level; total equity to total assets (TE/TA) ratio is used as an alternative measure of capital [13]. This approach allows us to better distinguish whether a bank's higher capitalization indicates its increased soundness, or whether it is merely a reflection of the higher risk it is facing. As the preliminary tests indicate relatively weak relationship between TE/TA ratio and independent variables, this measure was replaced with total equity to total liability (leverage) ratio in the follow up regressions.

### Independent and control variables

The choice of variables used in our analysis is primary guided by previous literature and data availability. These variables include both independent variables (risk measures and competition indicators) and set of control variables (institution and ownership indicators, profitability and efficiency ratios, liquidity, size, macroeconomic variables and dummy variables that capture year or country characteristics).

There are two main approaches to measuring bank competition: structural approach and non-structural approach [14]. As the name suggests, the structural approach assesses bank competition by examining measures of market structure such as concentration ratios (the share of assets held by the top 3 banking institutions) or indices (e.g., the Herfindahl-Hirschman index). The theoretical justification for using concentration as a measure of competition comes from the so called Structure-Conduct-Performance (SCP) paradigm, which postulates that fewer and larger firms (higher concentration) are more likely to engage in anticompetitive behavior [70]. The SCP hypothesis supports the notion that high concentrated firms are more competitive and profitable and have more market power in the framework of collusion. The

SCP examines the competition conditions by using ratios of concentration of largest firms and Herfindahl-Hirschman index (HHI) that characterize market structure. SCP paradigm is criticized on the assumption that causality is from structure to performance, though it is argued that conduct and performance can affect market structure.

In contrast to the structural approach, the non-structural approach is based on the so-called "New Empirical Industrial Organization literature", and measures competition without using explicit information about the structure of the market [13]. The non-structural measures focus on obtaining estimates of market power from the observed behavior of banks. For example [71], show that the sum of the elasticities of a firm's revenue with respect to the firm's input prices (the so-called H-Statistic) can be used to identify the extent of competition in a market. Various studies have used the H-Statistic to examine bank competition in different economic settings (see e.g., [13, 72–76] among others). [73] argue that the H-Statistic is a more appropriate measure for the degree of competition than other proxies for competitive conduct, and [77] notes that the H-Statistic is superior to other measures of competition, because it is derived from profit-maximizing equilibrium conditions.

This study uses Panzar-Rosse (PR) model which is an econometric approach in which competitive market conditions are to be assessed quantitatively. The model determines the competitiveness behavior of banks as per the comparative static features based on a reduced form of revenue equations using cross-section data [71, 72, 78]. Summing elasticity of the reduced form of revenues gives the so-called H-statistic, on which the model is based. The H-Statistic ranges from negative infinity ($-\infty$) to +1. The greater the value of H-statistic, the greater the competition is; a value of +1indicated perfect competition [79, 80], in which a bank's total revenue must change by the same percentage as its costs, and so, by the same percentage as its input prices. In the PR model it is assumed that banks have cost and revenue functions which allow to define profit maximization path, where marginal cost should be equal to the marginal revenue. Details on index definition and the estimation approach are provided in **S2 Appendix**. Following previous research on the MENA region [14], we explore the determinants of H-Statistic in the MENA region. The results are reported in (S3 Appendix) and show that the level of competition is determined by important institutional, regularity and bank-specific factors (see S2 Table in S2 Appendix).

We use the Herfindahl-Hirschman index (HHI) as another traditional measure of competition and concentration of the market conceived by [81, 82]. It is widely applied to estimate the level of competition of a market and its structure [83]. The HHI takes into account the relative size and the distribution of companies in a market and aims towards zero when the market consists of a large number of banks of relatively equal size. The more the value of the indicator increases, the more the market is concentrated, and weaker is the competition between the agents. The market thus aims towards a monopoly position and indicates an increase of the power of market. The decrease of the HHI indicates the opposite [63]. Details on index definition and the estimation approach are provided in **S2 Appendix**. We apply HHI in Eq (1) and (2) as a measure of banking concentration. Additionally, we create an interaction term between market competition indicator and the HH-index to investigate the impact of market competition on the relationship between banking concentration and different levels of capital and risk. There is an intensive research on banking competition in the MENA region that uses a variety of structural and non-structural measures (e.g., ratios of concentration, HH-index, PR-H statistic, and Lerner index) to examine the bank competitiveness and market power (see e.g., [1, 63, 76, 84]. However, we are the first to analyze the relationship between competition and capital ratios (as measure of financial soundness) of banks in the MENA region using the H-Statistic and the HH-Index. This allows us to provide some answers related to the role of

**Table 2. Descriptive statistics of sample banks.**

| | All Banks | | | | Conventional Banks | | | | Islamic Banks | | | | Mean test |
|---|---|---|---|---|---|---|---|---|---|---|---|---|---|
| | # obs | Mean | Median | SD | # obs | Mean | Median | SD | # obs | Mean | Median | SD | p-value |
| *Risk measures* | | | | | | | | | | | | | |
| Non-performing loans/ Total Loans (NPL/TL) | 1686 | 8.9% | 4.9% | 12.5% | 1281 | 8.9% | 5.0% | 12.9% | 405 | 9.2% | 4.4% | 12.9% | 0.534 |
| Loan loss provision/ Total Loans (LLP/TL) | 1785 | 7.6% | 4.4% | 10.9% | 1334 | 7.2% | 4.4% | 9.5% | 451 | 8.7% | 4.1% | 14.1% | 0.021** |
| Log Z | 2358 | 2.11 | 2.78 | 1.58 | 1774 | 2.26 | 2.90 | 1.56 | 584 | 1.65 | 2.34 | 1.56 | 0.000*** |
| *Capital Ratios* | | | | | | | | | | | | | |
| Ratio of total eligible capital to total assets (EC/TA) | 1820 | 21.5% | 14.6% | 37.5% | 1327 | 16.5% | 13.3% | 15.3% | 493 | 35.1% | 18.7% | 65.6% | 0.000*** |
| Ratio of total equity to total assets (TE/TA) | 2214 | 15.3% | 11.5% | 14.0% | 1647 | 13.1% | 11.2% | 8.8% | 567 | 21.8% | 12.7% | 22.2% | 0.000*** |
| *Market Competition* | | | | | | | | | | | | | |
| HH-index | 1392 | 0.21 | 0.18 | 0.13 | 442 | 0.20 | 0.16 | 0.13 | 1834 | 0.21 | 0.20 | 0.13 | 0.000*** |
| H-Statistics | 238 | 0.51 | 0.53 | 0.27 | 238 | 0.49 | 0.52 | 0.26 | 210 | 0.55 | 0.53 | 0.29 | 0.680 |
| Lerner index | 238 | 0.35 | 0.37 | 0.18 | 238 | 0.35 | 0.37 | 0.17 | 210 | 0.35 | 0.38 | 0.21 | 0.546 |
| *Profitability and Efficiency measures* | | | | | | | | | | | | | |
| Pre-tax income/ Total Assets (ROA) | 1823 | 1.5% | 1.5% | 2.4% | 1329 | 1.7% | 1.6% | 1.6% | 494 | 1.1% | 1.4% | 3.6% | 0.001*** |
| Cost-to-income ratio (CIR) | 2493 | 49.3% | 45.7% | 62.6% | 1864 | 47.4% | 45.0% | 45.6% | 629 | 54.8% | 47% | 96.7% | 0.347 |
| *Regulation and institution* | | | | | | | | | | | | | |
| *Activity Restrictions* | 238 | 4.20 | 5.00 | 3.08 | 238 | 4.31 | 5.00 | 3.04 | 210 | 3.93 | 5.00 | 3.16 | 0.004*** |
| *Institution* | 238 | -0.27 | -0.17 | 0.66 | 238 | -0.30 | -0.25 | 0.65 | 210 | -0.21 | -0.09 | 0.66 | 0.001*** |
| *Bank level characteristics* | | | | | | | | | | | | | |
| Deposit/ Total Assets | 2489 | 77.5% | 81.3% | 14.4% | 1880 | 79.3% | 81.5% | 10.6% | 609 | 72.0% | 80.1% | 21.8% | 0.000*** |
| Loan/Total Assets | 1779 | 49.9% | 54.5% | 19.8% | 1329 | 49.1% | 52.2% | 19.2% | 494 | 52.2% | 58.4% | 20.9% | 0.003*** |
| Loan/Total Deposits | 1792 | 67.0% | 69.0% | 44.2% | 1329 | 65.5% | 66.3% | 46.9% | 463 | 71.3% | 73.3% | 35.2% | 0.929 |
| Revenue Diversification (Ratio of non-interest revenue to TA) | 1817 | 2.3% | 1.2% | 4.4% | 1325 | 1.7% | 1.2% | 2.4% | 492 | 3.7% | 1.6% | 7.3% | 0.000*** |
| Leverage (Equity capital to Total Liability) | 1514 | 30.1% | 19.4% | 62.9% | 1179 | 24.7% | 18.7% | 36.9% | 335 | 49.0% | 22.2% | 112.4% | 0.015** |
| Size (Log(Assets)) | 2512 | 10.62 | 10.19 | 3.00 | 1863 | 10.69 | 10.24 | 2.85 | 649 | 10.41 | 10.14 | 3.38 | 0.417 |
| Ownership concentration | 1851 | 48.3% | 41.5% | 29.5% | 1366 | 49.6% | 43.2% | 29.5% | 485 | 44.6% | 39.7% | 29.4% | 0.000*** |
| Government ownership | 464 | 23.5% | 41.5% | 29.5% | 360 | 25.5% | 10.0% | 33.3% | 104 | 16.9% | 7.6% | 24.1% | 0.000*** |
| Foreign ownership | 1137 | 41.9% | 34.9% | 29.8% | 873 | 43.6% | 38.0% | 30.6% | 264 | 36.2% | 30.1% | 26.1% | 0.000*** |
| *Macroeconomic variables* | | | | | | | | | | | | | |
| GDP Growth | 238 | 4.0% | 3.5% | 4.0% | 238 | 3.9% | 3.4% | 3.8% | 210 | 4.0% | 3.7% | 4.4% | 0.941 |
| Inflation | 238 | -1.2% | -2.0% | 8.9% | 238 | 0.0% | -1.3% | 8.2% | 210 | -4.4% | -2.9% | 9.9% | 0.000*** |

The sample includes 225 banks in 18 countries in the MENA region. The sample of conventional banks includes 162 banks, and the sample of Islamic banks– 63 financial institutions. As measures of bank credit risk we use Loan loss reserves to gross loans (LLR/GL) and Non-performing loans to gross loans (NPL/GL), and for bank insolvency risk the measures are Distance-to-default (Z-score). Capital ratios are proxied by Total eligible capital to total assets (EC/TA) and Total equity to total assets (TE/TA). All the variables except regulation and institution are in percent. Bank characteristics for different group of banks are computed using data for the period 2006–2018. Bank-level characteristics, regulation, ownership, institution, and macroeconomic variables are described in S1 Appendix.

competitive conditions in the Islamic and conventional markets in the MENA region, and the impact of market structure on their financial stability.

According to [85], there are two types of ownership variables used to measure the internal corporate governance. These include concentration of ownership measured by the percentage of shareholding of the largest shareholder, and the types of ownership, that is, government shareholding and foreign ownership (see Table 2). In accordance with previous research [86, 87] we expect ownership concentration to have a negative effect on bank behavior, whereas government ownership should exert a positive effect. Likewise, we follow [19, 88] to predict that foreign ownership will reduce bank risk-taking in the MENA region. Our preliminary

tests show that only ownership concentration is statistically significant and exert a negative association with bank risk. Therefore, this variable is included in all regression models.

We follow [87, 89, 90], among others, in using several bank-specific characteristics known to be significant determinants of bank capital and risk. These include deposits, loans to assets ratio, loans to total deposits, total equity to total assets, pre-tax ROA and bank size, employed in the regression analysis as control variables (see Eq (1)). We follow [91] and employ cost-to-income ratio (CIR) as independent variable in our analysis of bank capital and risk. Specifically, we use CIR to control for differences in bank efficiency between the two groups of banks (Islamic and conventional). Following [92] approach, we create an index, *institution*, which is the mean of the six variables for each country in the sample. A higher value of the index indicates better institutional environment in the sample country. Finally, we use the GDP growth rate, GDP per capital (as alternative measure), and inflation to control for macroeconomic differences across the countries in our sample [2, 13].

## Results and discussions

### Descriptive statistics and univariate analysis

Table 2 compares different variables used in our analysis across the two banking systems (CBs and IBs). We use two alternative measures for a' bank capitalization level (EC/TA and TE/TL); similarity, we use HH-index and H-Statistic as alternative measures of market competition (**see S1 Data**).

We observe a statistically significant difference between CBs and IBs in all measures of bank risk (the mean difference is significant at the usual levels of significance) except for non-performing loans to gross loans ratio. The two alternative measures of the capitalization ratio (EC/TA and TE/TA) are also statistically different between the two groups of banks at the 1% level of significance. The estimated value of profitability ratio (pre-tax ROA) shows that CBs experience a better performance over the sample period of fourteen years than IBs (1.70% vs 1.13%). Our results are in line with [93] who find that during and after the global financial crisis in 2007–2008, the IBs performance has significantly deteriorated. Furthermore, the data in Table 2 shows that IBs have higher level of inefficiency measured by cost-to-income ratio (CIR) than CBs (54.82% vs. 47.41%); however, the mean difference between the two samples is statistically insignificant.

According to Table 2, we observe that capital ratio (EC/TA) mean value for the two groups of banks (Islamic and conventional) is between 17% and 35%, which is well above the minimum capitalization required by the Basel agreements. The results are similar for TE/TA ratio. These results are in line with previous research on Islamic banking which indicates that IBs are more stable compared to their conventional counterparts, and this is due to the strong capital ratios that were considerably higher than those of the traditional banks [2, 66]. However, a study by [57] reveals that though the average capitalization ratio of IBs is higher, there is no significant difference in capitalization level between the two banking systems. Our data analysis does not support this evidence; we observe a statistically significant difference in the capital ratios between CBs and IBs. Table 2 provide also data for the Lerner index for the respective countries included in the sample. Further, we do not observe a significant difference in the level of banking competition (measured by the H-Statistic) between the two samples. The overall HH-index is 0.21, which is considered "moderately concentrated" for all countries in the sample, with a mean difference strongly significant at the 1% level of significance.

Next, we compare the individual bank-level characteristics between the two samples, and find that the bank-specific variables are significantly different between CBs and IBs (except net loans to total assets). Our results reported in Table 2 support the findings of previous research

on banks in the MENA region [93–95]. According to [96], Islamic profit-loss sharing products present greater insolvency risk than products offered by CBs, and this type of risk has a more detrimental impact on bank performance during a prolonged crisis. Table 2 confirms this finding; the insolvency risk measured by distance-to-default (or Z-score) is significantly different between the two groups of banks, with CBs much less risky than IBs (2.26 vs. 1.65). We also observe that IBs have higher credit risk too.

Finally, the analysis shows that CBs have higher percentage of ownership concentration than IBs (49.62% vs. 44.65%). The high percentage of concentrated ownership in both types of banks is in line with the previous studies' observations for the MENA region [94, 97]. Furthermore, around 24% of all banks in the sample are government-owned with IBs having a lower percentage of government ownership. Following the approach of [92], we create a composite index, *institution*, which is the mean of six variables for each country in the sample; a higher value of the index indicates better institutions. In line with previous studies on the MENA region, we find that the countries in our sample are characterized with week institutional environment (a median value of -0.17 for the total sample). Activity restriction variable is also statistically different between IBs and CBs which indicates that the impact of regulatory restrictions can be different between the two baking systems.

## The impact of market competition and credit risk on capital ratios

In line with our first hypothesis (H1a), we expect market competition and risk-taking to have a significant impact on capitalization level of banks in the MENA region. Furthermore, we hypothesize that banks raise their capitalization levels in response to a higher risk rather than the other way round. We run our analysis using two alternative ratios (EC/TA and TE/TL) as a proxy for a bank's capitalization level (**see S1 File**). The outputs of the regression analysis are reported in Table 3.

First, we consider the results without estimating the effect of market competition (see Model 1). In line with the regulatory hypothesis, in which capital and risk are positively associated [69], we find that credit risk exerts a positive influence on banks capitalization level; the positive relationship indicates that if banks raises the credit risk by 1%, then capital level will increase by 0.524%. We also run the regressions with Z-score (or distance to default) as a measure of a bank's insolvency risk and find that the greater financial stability (high Z-score) of bank promotes capital ratio which is in line with [90, 98]. The relationship between the index of activity restriction and capital ratios is negative yet insignificant; therefore, we do not find evidence to support the notion that less activity restrictions are associated with increased banking stability. We also test the hypothesis that the effect of increased competition on capital ratios may be larger in magnitude in countries with a higher proportion of non-performing loans since bank charter values will suffer [13]. Therefore, in the next two models, we introduce the H-Statistic and the HH-index as measures of market competition and concentration, respectively. The HHI enters all regressions in Table 3 positively and significantly, indicating that banks hold more capital when concentration increases. The coefficient of H-Statistic is however, statistically insignificant and negative, which contradicts the findings of previous research that shows a positive association (see [1] for the MENA region banks and [13] for European banks).

However, we observe a positive impact of market completion (measured by the H-Statistic) on capital ratio when TE/TL is used as a dependent variable in our models. The positive association between the two variables suggests prudent behavior on the part of the banks when competition strengthens. Moreover, this result is in line with the predictions of theoretical studies by [46, 47]. On the other hand, the positive sign of the HHI variable supports the predictions

**Table 3. Panel regressions of competition and capital ratio (all banks, 2005–2018).**

| Explanatory Variables | EC/TA | | | | TE/TL | | | |
|---|---|---|---|---|---|---|---|---|
| | Model 1 | Model 2 | Model 3 | Model 4 | Model 5 | Model 6 | Model 7 | Model 8 |
| Constant | -0.110 | -0.136 | -0.091 | -0.115 | -0.095 | -0.118 | -0.149 | -0.174 |
| | (0.349) | (0.246) | (0.442) | (0.331) | (0.603) | (0.519) | (0.417) | (0.345) |
| Credit risk | 0.524*** | 0.497*** | 0.520*** | 0.490*** | 0.895*** | 0.869*** | 0.902*** | 0.876*** |
| | (0.000) | (0.000) | (0.000) | (0.000) | (0.000) | (0.000) | (0.000) | (0.000) |
| HHI | | 0.144*** | | 0.144*** | | 0.112* | | 0.113* |
| | | (0.002) | | (0.002) | | (0.103) | | (0.101) |
| H-Stat | | | -0.039 | -0.037 | | | 0.109*** | 0.110*** |
| | | | (0.122) | (0.136) | | | (0.005) | (0.005) |
| | | | | -0.022 | | | | 0.007 |
| HHI* H-Stat | | | | (0.125) | | | | (0.756) |
| Activity index | -0.001 | -0.009 | -0.007 | -0.004 | 0.007*** | 0.007*** | 0.006** | 0.006** |
| | (0.539) | (0.589) | (0.672) | (0.796) | (0.006) | (0.005) | (0.016) | (0.015) |
| Ownership concentration | 0.008 | 0.010 | 0.008 | 0.010 | 0.013 | 0.012 | 0.012 | 0.012 |
| | (0.611) | (0.542) | (0.608) | (0.534) | (0.621) | (0.629) | (0.640) | (0.650) |
| Institution | -0.020* | -0.019* | -0.021* | -0.019* | -0.115*** | -0.114*** | -0.113*** | -0.112*** |
| | (0.251) | (0.284) | (0.241) | (0.284) | (0.000) | (0.000) | (0.000) | (0.000) |
| Cost-income ratio | -0.007 | -0.007 | -0.007 | -0.006 | -0.002** | -0.002** | -0.002*** | -0.002** |
| | (0.164) | (0.199) | (0.184) | (0.225) | (0.012) | (0.015) | (0.009) | (0.011) |
| Deposit/Total Assets | -0.105*** | -0.113*** | -0.106*** | -0.114*** | 0.064 | 0.051 | 0.063 | 0.050 |
| | (0.000) | (0.000) | (0.000) | (0.000) | (0.110) | (0.211) | (0.115) | (0.221) |
| Loan/ Total Assets | 0.237*** | 0.202*** | 0.241*** | 0.208*** | -0.245*** | -0.237*** | -0.244*** | -0.236*** |
| | (0.000) | (0.000) | (0.000) | (0.000) | (0.000) | (0.000) | (0.000) | (0.000) |
| Loan/Total Deposit | 0.020 | 0.020 | 0.019 | 0.019 | 0.194*** | 0.183*** | 0.192*** | 0.181*** |
| | (0.299) | (0.287) | (0.323) | (0.329) | (0.000) | (0.000) | (0.000) | (0.000) |
| Revenue Diversification | 0.657*** | 0.654*** | 0.664*** | 0.669*** | -0.384 | -0.375 | -0.399 | -0.393 |
| | (0.000) | (0.000) | (0.000) | (0.000) | (0.143) | (0.153) | (0.128) | (0.134) |
| Equity/TL (1–4) Equity/TA (5–8) | 0.128*** | 0.126*** | 0.129*** | 0.127*** | 0.103* | 0.104* | 0.102* | 0.103* |
| | (0.000) | (0.000) | (0.000) | (0.000) | (0.084) | (0.082) | (0.085) | (0.082) |
| Pre-tax ROA | -1.491*** | -1.579*** | -1.494*** | -1.589*** | 1.615*** | 1.504*** | 1.601*** | 1.491*** |
| | (0.000) | (0.000) | (0.000) | (0.000) | (0.000) | (0.001) | (0.000) | (0.001) |
| Size | 0.001 | 0.001 | 0.001 | 0.002 | 0.006 | 0.007 | 0.003 | 0.004 |
| | (0.506) | (0.378) | (0.485) | (0.355) | (0.854) | (0.819) | (0.919) | (0.887) |
| GDP Growth | 0.263* | 0.261* | 0.210* | 0.183* | 0.025 | 0.037 | 0.178 | 0.200 |
| | (0.092) | (0.094) | (0.088) | (0.053) | (0.915) | (0.875) | (0.472) | (0.421) |
| Inflation | -0.133 | -0.135 | -0.129 | -0.144 | 0.292* | 0.289* | 0.280* | 0.281* |
| | (0.167) | (0.159) | (0.180) | (0.135) | (0.051) | (0.053) | (0.062) | (0.061) |
| ISLAMIC_D | 0.083*** | 0.085*** | 0.083*** | 0.085*** | 0.083*** | 0.085*** | 0.083*** | 0.084*** |
| | (0.000) | (0.000) | (0.000) | (0.000) | (0.000) | (0.000) | (0.000) | (0.000) |
| CRISIS_D | -0.031* | 0.016* | -0.033** | -0.034** | -0.030 | -0.030 | -0.023 | -0.022 |
| | (0.062) | (0.051) | (0.045) | (0.037) | (0.239) | (0.244) | (0.373) | (0.382) |
| Country Dummy | Yes | Yes | Yes | Yes | Yes | Yes | Yes | Yes |
| Year Dummy | Yes | Yes | Yes | Yes | Yes | Yes | Yes | Yes |
| Number of Observations | 2901 | 2901 | 2901 | 2901 | 2901 | 2901 | 2901 | 2901 |

(*Continued*)

**Table 3.** (Continued)

| Explanatory Variables | EC/TA | | | | TE/TL | | | |
|---|---|---|---|---|---|---|---|---|
| | Model 1 | Model 2 | Model 3 | Model 4 | Model 5 | Model 6 | Model 7 | Model 8 |
| R-squared (Overall) | 0.2284 | 0.2309 | 0.2291 | 0.2321 | 0.1941 | 0.1949 | 0.1963 | 0.1971 |

The panel data regressions estimate the relation between banking competition and capital ratio over the period of 2005–2018 while controlling for important bank-level and macroeconomic characteristics. The sample includes 225 banks in 18 countries in the MENA region. Banks included in the sample are conventional banks (162) and Islamic banks (63). As a measure of bank capitalization level we use the Total eligible capital to total assets (EC/TA) and Total equity to total liability (TE/TL). Bank-level characteristics and capital ratio are computed as of year $t-1$. All the regressions control for year and country fixed effects.

*, **, and *** indicate statistical significance at the 10%, 5%, and 1% level, respectively. The values in parenthesis represent 'p-value'. Capital adequacy ratio, market competition indicators, bank-level characteristics, institution, ownership and macroeconomic variables are described in S1 Appendix.

of the Structure-Conduct-Performance (SCP) hypothesis which postulates that higher concentration would lead to less competition and, consequently, greater financial stability (in our case banks' capital ratios are used as a measure of soundness). This result is also in line with our first hypothesis (H1a). The estimated coefficient of the interaction term between HHI and H-Statistic introduced in Model 4 is negative yet insignificant; therefore, we cannot confirm the notion that increased concentration does have to be associated with uncompetitive markets. For our sample of MENA banks, this means that an increase in banking competition in countries with an average moderate level will not enhance the positive impact of market concentration on banks' capitalization level.

Following [95], we introduce in each model a composite variable, *institution*, which measures the overall quality of institutional environment in the sample countries. We find that this variable is marginally significant and negative in all the regressions. The policy implication of this finding would be that an improved institutional environment in the MENA countries combined with sound prudential regulation will prevent banks from increasing their capital over minimum requirements which in term may induce banks to take more risk. Previous research finds a significant impact of ownership structure on the regulatory capital and risk behavior of banks in the MENA region. For example [99], finds an inverse association between ownership concentration and bank risk-taking in the MENA countries. Our results for ownership variable does not support this finding; the estimated coefficient of ownership concentration is insignificant in all models.

Our main results continue to hold after controlling for a number of common bank-level determinants of risk and capital; all of them (except Loans/Total Deposits ratio and Cost-to-Income ratio) have a strong influence on the capitalization level of banks. Most of these variables hold signs and magnitude as predicted in the empirical literature. For example, in line with [32], we find a strong positive association of capital ratios with loans (as percentage of total assets or deposits), revenue diversification, and leverage. However, we are not able to provide evidence that banks with different size will hold different level of capital (the estimated coefficient of the size variable is insignificant in all regressions). In line with previous research (see e.g., [2]), we find that macroeconomic conditions in the MENA region (more specifically, GDP growth rate and inflation) have a strong impact on the level of bank capitalization. Further, we find that during the global financial crisis of 2008–2009, banks tend to keep lower capital ratios than during the non-crisis period. As IB dummy variable is strongly significant in all the regressions, we hypothesize that the effect of market competition and risk-taking on banks' capital ratios can be different between the two banking systems. We investigate this issue in-death in the next section.

The results reported in Model 5 to 8 using an alternative measure of bank capitalization level (TE/TL) provide further support to our findings. However, few significant differences are

**Table 4. Panel regressions of competition and capital ratio (CBs, 2006–2018).**

| Explanatory Variables | EC/TA | | | | TE/TL | | | |
|---|---|---|---|---|---|---|---|---|
| | Model 1 | Model 2 | Model 3 | Model 4 | Model 5 | Model 6 | Model 7 | Model 8 |
| Constant | -0.024 | -0.052 | -0.020 | -0.063 | -0.010 | -0.051 | -0.029 | -0.071 |
| | (0.678) | (0.366) | (0.725) | (0.283) | (0.949) | (0.750) | (0.856) | (0.662) |
| Credit risk | 0.224*** | 0.207*** | 0.215*** | 0.204*** | 0.569*** | 0.537*** | 0.570*** | 0.537*** |
| | (0.000) | (0.000) | (0.000) | (0.000) | (0.000) | (0.000) | (0.000) | (0.000) |
| HHI | | 0.069*** | | 0.070*** | | 0.102** | | 0.103** |
| | | (0.000) | | (0.000) | | (0.048) | | (0.046) |
| H-Stat | | | 0.016 | 0.023** | | | 0.033 | 0.031 |
| | | | (0.150) | (0.038) | | | (0.287) | (0.274) |
| HHI*H-Stat | | | | -0.015** | | | | -0.008 |
| | | | | (0.011) | | | | (0.961) |
| Activity index | 0.001*** | 0.002*** | 0.002*** | 0.001*** | 0.004** | 0.004** | 0.004** | 0.004** |
| | (0.007) | (0.005) | (0.002) | (0.008) | (0.025) | (0.021) | (0.037) | (0.032) |
| Ownership concentration | 0.023*** | 0.024*** | 0.021*** | 0.024*** | 0.030 | 0.030 | 0.030 | 0.030 |
| | (0.001) | (0.001) | (0.003) | (0.001) | (0.121) | (0.124) | (0.125) | (0.128) |
| Institution | -0.014* | -0.014* | -0.012* | -0.012* | -0.037* | -0.036* | -0.035* | -0.034* |
| | (0.052) | (0.056) | (0.098) | (0.088) | (0.070) | (0.076) | (0.087) | (0.096) |
| Cost-income ratio | -0.002 | -0.002 | -0.002 | -0.002 | -0.001** | -0.001** | -0.001** | -0.001** |
| | (0.233) | (0.302) | (0.284) | 0.287 | (0.019) | (0.027) | (0.018) | (0.025) |
| Deposit/Total Assets | -0.023** | -0.026*** | -0.035*** | -0.026*** | 0.047 | 0.037 | 0.047 | 0.036 |
| | (0.014) | (0.005) | (0.000) | (0.005) | (0.157) | (0.279) | (0.160) | (0.284) |
| Loan/ Total Assets | 0.167*** | 0.147*** | 0.148*** | 0.150*** | -0.177*** | -0.169*** | -0.175*** | -0.168*** |
| | (0.000) | (0.000) | (0.000) | (0.000) | (0.000) | (0.001) | (0.000) | (0.001) |
| Loan/Total Deposit | 0.042*** | 0.046*** | 0.043*** | 0.042*** | 0.137*** | -0.129*** | 0.135*** | 0.126*** |
| | (0.000) | (0.000) | (0.000) | (0.000) | (0.000) | (0.000) | (0.000) | (0.000) |
| Revenue Diversification | -0.080 | -0.139 | -0.107 | -0.061 | -2.503*** | -2.463*** | -2.430*** | -2.387*** |
| | (0.665) | (0.451) | (0.562) | (0.742) | (0.000) | (0.000) | (0.000) | (0.000) |
| Equity/TL (1–4) Equity/TA (5–8) | 0.164*** | 0.162*** | 0.162*** | 0.162*** | 0.032 | 0.029 | 0.033 | 0.030 |
| | (0.000) | (0.000) | (0.000) | (0.000) | (0.616) | (0.646) | (0.604) | (0.636) |
| Pre-tax ROA | -0.004 | -0.053 | -0.078 | -0.103 | 4.071*** | 3.880*** | 4.025*** | 3.830*** |
| | (0.981) | (0.779) | (0.680) | (0.586) | (0.000) | (0.000) | (0.000) | (0.000) |
| Size | -0.002*** | -0.002*** | -0.001*** | -0.002*** | -0.001 | -0.008 | -0.001 | -0.008 |
| | (0.002) | (0.004) | (0.003) | (0.003) | (0.684) | (0.736) | (0.674) | (0.727) |
| GDP Growth | 0.078 | 0.076 | 0.166** | 0.085 | -0.124 | -0.111 | -0.081 | -0.068 |
| | (0.263) | (0.274) | (0.029) | (0.233) | (0.519) | (0.562) | (0.681) | (0.731) |
| Inflation | 0.033 | 0.035 | 0.077 | 0.015 | -0.013 | -0.009 | -0.025 | -0.022 |
| | (0.453) | (0.431) | (0.131) | (0.729) | (0.916) | (0.937) | (0.842) | (0.857) |
| CRISIS_D | -0.018*** | -0.019*** | -0.037** | -0.017** | -0.036* | -0.036* | -0.035* | -0.034* |
| | (0.006) | (0.006) | (0.015) | (0.010) | (0.052) | (0.057) | (0.064) | (0.071) |
| Country Dummy | Yes | Yes | Yes | Yes | Yes | Yes | Yes | Yes |
| Year Dummy | Yes | Yes | Yes | Yes | Yes | Yes | Yes | Yes |
| Number of Observations | 2106 | 2106 | 2106 | 2106 | 2106 | 2106 | 2106 | 2106 |
| R-squared (Overall) | 0.5515 | 0.5541 | 0.5486 | 0.5565 | 0.2059 | 0.2073 | 0.2063 | 0.2078 |

The panel data regressions estimate the relation between banking competition and capital ratios over the period of 2006–2018 while controlling for important bank-level and macroeconomic characteristics. The sample includes 225 banks in 18 countries in the MENA region. Banks included in the sample are only conventional banks. As a measure of bank capitalization level we use the Total eligible capital to total assets (EC/TA) and Total equity to total liability (TE/TL) ratios. All the regressions control for year and country fixed effects.

*, **, and *** indicate statistical significance at the 10%, 5%, and 1% level, respectively. The values in parenthesis represent 'p-value'. Capital adequacy ratio, market competition indicators, bank-level characteristics, institution, ownership and macroeconomic variables are described in S1 Appendix.

observed. First, activity restrictions are known to be a key determinant for the scope of a bank's business. A growing body of empirical evidence [7, 13] suggests that increased competition, increased concentration and sectors with greater contestability and less activity restrictions, are all associated with banking stability. In line with this notion, we find that banks in the MENA region increase their capitalization level to cope with the increased restrictions on the side of regulators. Second, in line with [1], the H-Statistic is positively and significantly associated with capital ratios, indicating that banks increase their capital levels when competition increases. This finding does corroborate the evidence of previous research concerning the positive relation of the H-Statistic with capital ratios. Thus, our results empirically substantiate that competition is positively linked with bank soundness. Our findings have strong implication for bank managers who need to keep a sufficient capital level to limit the impact of downside risk from depletion of capital buffers which is perceived to be significant during the COVID-19 pandemic.

## The differential effect of competition on Islamic banks

The results reported in Table 3 do not provide clear answer to the question of whether the impact of competition and credit risk on bank capital ratios is significantly different between CBs and IBs. To the best of our knowledge this is one of the rare studies to have addressed this important question with strong policy implications. It is also worth investigating the reasons for such differences. Therefore, we run our analysis separately for the samples of CBs and IBs. The outputs of the regression analysis are reported in Tables 4 and 5 (**see S2 and S3 Files**).

In Table 4, we report the results for CBs sample using EC/TA ratio as a measure of bank capitalization level. We first consider the results without estimating the effect of market competition (see Model 1). In line with the regulatory hypothesis, we find that credit risk exerts a strong positive influence on bank capital; this effect is statistically significant at the 1% level of significance. The relationship between the index of activity restriction and capital ratios is positive and strongly significant in all the regressions; this contradicts the general notion that the increase in activity restrictions on bank activities will reduce the banks' capitalization level that leads to decrease in their financial soundness. In the next two models, we introduce the HH-index and the H-Statistic as measures of market competition and concentration, respectively. We find strong evidence that CBs raise their capital level when concentration increases, which is in line with previous research [1, 13]. Individually, the coefficient of H-Statistic is statistically insignificant yet positive. The analysis of the moderating effect of market competition (see Model 4) indicates that the increased level of market competition reduces the positive impact of concentration on the capitalization level of CBs. In other words, increased competition will restrict the more concentrated banks from increasing their capitalization level and risk. This result contradicts empirical studies which obtained positive relationship between competition and concentration [1, 74], and calls into question the strategy taken by policy makers in some MENA countries to increase the level of bank concentration in order to improve financial stability.

The results in Table 5 indicates a differential effect of competition on capital ratios of IBs. For example, in both samples we observe a positive association between market concentration and capital; however, this effect is much more pronounced in the sample of IBs, which provides support to our second hypothesis (H2a). Additionally, the effect of market competition (measured by the H-Statistics) is strongly significant and negative only in the sample of IBs. This finding suggests that, in general, Islamic banking institutions do not increase their capitalization level in face of increased competition. Since competition in the banking market primarily affects the interest rate, it can be concluded that IBs are about to apply their theoretical

**Table 5. Panel regressions of competition and capital ratio (IBs, 2006–2018).**

| Explanatory Variables | EC/TA | | | | TE/TL | | | |
|---|---|---|---|---|---|---|---|---|
| | Model 1 | Model 2 | Model 3 | Model 4 | Model 5 | Model 6 | Model 7 | Model 8 |
| Constant | -0.224 | -0.193 | -0.149 | -0.120 | -0.357 | -0.367 | -0.439 | -0.447 |
| | (0.459) | (0.524) | (0.622) | (0.691) | (0.365) | (0.352) | (0.265) | (0.258) |
| Credit risk) | 0.958*** | 0.922*** | 0.936*** | 0.902*** | 1.565*** | 1.591*** | 1.565*** | 1.588*** |
| | (0.000) | (0.000) | (0.000) | (0.000) | (0.000) | (0.000) | (0.000) | (0.000) |
| HHI | | 0.383** | | 0.359** | | -0.181 | | -0.163 |
| | | (0.033) | | (0.046) | | (0.423) | | (0.472) |
| H-Stat | | | -0.205*** | -0.195** | | | -0.225** | -0.222** |
| | | | (0.008) | (0.012) | | | (0.024) | (0.026) |
| HHI*H-Stat | | | | -0.038 | | | | 0.001 |
| | | | | (0.408) | | | | (0.979) |
| Activity index | -0.009 | -0.010 | -0.007 | -0.007 | 0.018** | 0.019** | 0.016* | 0.016* |
| | (0.139) | (0.125) | (0.238) | (0.249) | (0.027) | (0.025) | (0.055) | (0.053) |
| Ownership concentration | -0.033 | -0.021 | -0.043 | -0.032 | -0.046 | -0.046 | -0.037 | -0.038 |
| | (0.618) | (0.748) | (0.509) | (0.629) | (0.588) | (0.587) | (0.658) | (0.657) |
| Institution | -0.039 | -0.030 | -0.015 | -0.006 | -0.371*** | -0.374** | -0.396*** | -0.398*** |
| | (0.534) | (0.628) | (0.811) | (0.917) | (0.000) | (0.000) | (0.000) | (0.000) |
| Cost-income ratio | -0.001 | 0.001 | 0.001 | 0.003 | -0.038 | -0.040 | -0.039 | -0.041 |
| | (0.956) | (0.954) | (0.963) | (0.876) | (0.162) | (0.141) | (0.154) | (0.136) |
| Deposit/Total Assets | -0.247*** | -0.267*** | -0.254*** | -0.274*** | -0.118 | -0.101 | -0.109 | -0.094 |
| | (0.000) | (0.000) | (0.000) | (0.000) | (0.263) | (0.345) | (0.297) | (0.377) |
| Loan/ Total Assets | 0.273** | 0.201** | 0.310** | 0.242* | -0.276* | -0.280** | -0.290** | -0.293** |
| | (0.028) | (0.018) | (0.013) | (0.061) | (0.051) | (0.048) | (0.040) | (0.039) |
| Loan/Total Deposit | 0.098 | 0.074 | 0.088 | 0.066 | 0.552*** | 0.578*** | 0.541*** | 0.565*** |
| | (0.201) | (0.335) | (0.247) | (0.389) | (0.000) | (0.000) | (0.000) | (0.000) |
| Revenue Diversification | 0.681** | 0.660** | 0.774** | 0.755** | -0.151 | -0.136 | -0.261 | -0.247 |
| | (0.044) | (0.050) | (0.022) | (0.025) | (0.731) | (0.758) | (0.555) | (0.578) |
| Equity/TL (1–4) Equity/TA (5–8) | 0.089*** | 0.087*** | 0.097*** | 0.095*** | 0.164 | 0.158 | 0.165 | 0.160 |
| | (0.002) | (0.002) | (0.001) | (0.001) | (0.147) | (0.164) | (0.143) | (0.158) |
| Pre-tax ROA | -1.523** | -1.631** | -1.598** | -1.700** | 0.471 | 0.555 | 0.519 | 0.594 |
| | (0.024) | (0.016) | (0.017) | (0.011) | (0.586) | (0.525) | (0.547) | (0.494) |
| Size | 0.015* | 0.017** | 0.016* | 0.017** | 0.016* | 0.016* | 0.016* | 0.015* |
| | (0.061) | (0.038) | (0.051) | (0.031) | (0.129) | (0.142) | (0.146) | (0.159) |
| GDP Growth | 0.499 | 0.475 | 0.222 | 0.175 | 0.275 | 0.270 | 0.595 | 0.589 |
| | (0.300) | (0.323) | (0.650) | (0.722) | (0.657) | (0.662) | (0.348) | (0.356) |
| Inflation | -0.404* | -0.422* | -0.505* | -0.533* | 0.808** | 0.814** | 0.914** | 0.919** |
| | (0.108) | (0.100) | (0.072) | (0.058) | (0.027) | (0.026) | (0.013) | (0.013) |
| CRISIS_D | 0.052 | 0.068 | 0.071 | 0.085 | 0.061 | 0.067 | 0.084 | 0.013 |
| | (0.369) | (0.244) | (0.227) | (0.150) | (0.416) | (0.377) | (0.268) | (0.244) |
| Country Dummy | Yes | Yes | Yes | Yes | Yes | Yes | Yes | Yes |
| Year Dummy | Yes | Yes | Yes | Yes | Yes | Yes | Yes | Yes |
| Number of Observations | 795 | 795 | 795 | 795 | 795 | 795 | 795 | 795 |
| R-squared (Overall) | 0.2035 | 0.2082 | 0.2108 | 0.2154 | 0.3281 | 0.3287 | 0.3326 | 0.3330 |

The panel data regressions estimate the relation between banking competition and capital ratios over the period of 2006–2018 while controlling for important bank-level and macroeconomic characteristics. The sample includes 225 banks in 18 countries in the MENA region. Banks included in the sample are only Islamic banks. As a measure of bank capitalization level we use the Total eligible capital to total assets (EC/TA) and Total equity to total liability (TE/TL) ratios. All the regressions control for year and country fixed effects.

*, **, and *** indicate statistical significance at the 10%, 5%, and 1% level, respectively. The values in parenthesis represent 'p-value'. Capital adequacy ratio, market competition indicators, bank-level characteristics, institution, ownership and macroeconomic variables are described in S1 Appendix.

model based on the prohibition of interest. However, we expect that market competition will have a positive effect on the relationship between banking concentration and capital ratios, that is, more concentrated banks will hold higher capital ratios if they have to deal with increased competition. Our analysis does not support this notion as the estimated coefficient of the interactive term in Model 4 is insignificant. Therefore, we should expect that IBs will behave differently than CBs under competitive conditions, which contradicts our second hypothesis (H2b). The main reason lies in different business model and risk management practices employed by these banks.

Our findings have important policy implications for the banking sector during the COVID-19 pandemic. More specifically, compared to conventional banking institutions, Islamic banks are more dynamic in upholding their capital level during the crisis period though in normal economic conditions the capitalization levels of conventional banks are higher than those of IBs. This finding is confirmed by [30] who report that Islamic banks are more efficient to hold capital at a higher level than those of conventional (and state-owned) banks during the COVID-19 pandemic.

Our main results continue to hold after controlling for a number of common bank-level determinants of capital and risk which demonstrate a strong influence on the capitalization level of either bank. Further, we find that during the global financial crisis of 2008–2009, CBs tend to keep lower capitalization level than during the non-crisis period. This variable is, however, insignificant in the sample of IBs. Next, we repeat our analysis in Tables 4 and 5 using an alternative measure of banks' capitalization level–Total Equity to Total Liabilities (TE/TL). The estimation results are not quite different from our previous findings reported in the same tables. The analysis reveals significant differences in the competition effect for CBs and IBs and their capitalization levels. Thus, our findings inform regulatory authorities concerned with improving the financial stability of banking sector in the MENA region to strengthen their policies (in this case capital requirements) in order to force banks to better align with the strengthen capital requirements during the COVID-19 pandemic. This policy implication is especially important for the MENA countries with moderate level of competition since the increase of competition will lead to an increase of financial stability of the whole banking system.

## The impact of market competition and capital ratios on credit risk

Previous research has reached the conclusion that banks raise their capitalization levels in response to a higher risk rather than the other way round [30]. The results in Table 3 confirm the positive association between capital ratios and the level of credit risk of banks in the MENA region. We also test the opposite hypothesis–banks raise their risk in a response to the need to increase their capitalization level. We run our analysis using two alternative measures of credit risk—LLR/GL and NPL/GL, respectively (**see S1 File**). The outputs of the regression analysis are reported in Table 6.

In line with our fist hypothesis (H1b), according to which competition and capital ratios have a significant impact on bank risk, we find that capital requirements exert a positive influence on bank credit risk (see Model 1). This finding is well documented in the empirical literate on the MENA region [2] but contradicts [29] who finds a significant negative association between the two variables for Bangladeshi banks. In the next two models, we introduce the HH-index and the H-Statistic as measures of market competition and concertation, respectively. HH-index enters the regressions in Table Table 6 positively and significantly (see Model 2), The positive linear relationship between HH-index and credit risk indicates that an increase in banking concentration (in order to lessen the competition) leads to a reduction in the level

**Table 6. Panel regressions of competition and credit risk (all banks, 2006–2018).**

| | LLR/GL | | | | NPL/GL | | | |
|---|---|---|---|---|---|---|---|---|
| Explanatory Variables | Model 1 | Model 2 | Model 3 | Model 4 | Model 5 | Model 6 | Model 7 | Model 8 |
| Constant | -0.033 | -0.061 | -0.015 | -0.043 | -0.047 | -0.066 | -0.037 | -0.056 |
| | (0.376) | (0.100) | (0.689) | (0.250) | (0.173) | (0.153) | (0.279) | (0.105) |
| EC/TA | 0.033*** | 0.170*** | 0.032*** | 0.026*** | 0.049*** | 0.045*** | 0.049*** | 0.044*** |
| | (0.000) | (0.000) | (0.000) | (0.000) | (0.000) | (0.000) | (0.000) | (0.000) |
| HHI | | 0.144*** | | 0.169*** | | 0.102*** | | 0.102*** |
| | | (0.002) | | (0.000) | | (0.000) | | (0.000) |
| H-Stat | | | -0.039*** | -0.036*** | | | -0.020*** | -0.019** |
| | | | (0.000) | (0.000) | | | (0.008) | (0.010) |
| HHI*H-Stat | | | | -0.006 | | | | -0.010** |
| | | | | (0.162) | | | | (0.013) |
| Activity index | 0.001*** | 0.001***s | 0.001*** | 0.002*** | 0.001** | 0.001** | 0.001** | 0.001*** |
| | (0.007) | (0.003) | (0.001) | (0.000) | (0.037) | (0.028) | (0.015) | (0.007) |
| Ownership concentration | -0.010* | -0.008* | -0.010* | -0.009* | -0.005 | -0.005 | -0.005 | -0.005 |
| | (0.061) | (0.103) | (0.051) | (0.098) | (0.265) | (0.269) | (0.250) | (0.257) |
| Institution | -0.012** | -0.010* | -0.013** | -0.011* | -0.010* | -0.009* | -0.010* | -0.009* |
| | (0.038) | (0.078) | (0.026) | (0.059) | (0.064) | (0.088) | (0.054) | (0.080) |
| Cost-income ratio | 0.006*** | 0.006*** | 0.001*** | 0.007*** | 0.001*** | 0.001*** | 0.001*** | 0.001*** |
| | (0.000) | (0.000) | (0.000) | (0.000) | (0.000) | (0.000) | (0.000) | (0.000) |
| Deposit/Total Assets | 0.019*** | 0.007* | 0.019*** | 0.007* | 0.041*** | 0.027*** | 0.041*** | 0.027*** |
| | (0.008) | (0.094) | (0.008) | (0.090) | (0.000) | (0.001) | (0.000) | (0.001) |
| Loan/ Total Assets | 0.018 | -0.024 | 0.019* | -0.021* | -0.056*** | -0.048*** | -0.056*** | -0.047*** |
| | (0.139) | (0.053) | (0.107) | (0.078) | (0.000) | (0.000) | (0.000) | (0.000) |
| Loan/Total Deposit | -0.031*** | -0.029*** | -0.031*** | -0.030*** | -0.026*** | -0.035*** | -0.026*** | -0.035*** |
| | (0.000) | (0.000) | (0.000) | (0.000) | (0.000) | (0.000) | (0.000) | (0.000) |
| Revenue Diversification | 0.383*** | 0.365*** | 0.386*** | 0.370*** | 0.469*** | 0.469*** | 0.471*** | 0.473*** |
| | (0.000) | (0.000) | (0.000) | (0.000) | (0.000) | (0.000) | (0.000) | (0.000) |
| Pre-tax ROA | 0.003 | -0.110 | 0.002 | -0.113 | -0.037 | -0.139* | -0.036 | -0.143* |
| | (0.972) | (0.230) | (0.977) | (0.220) | (0.658) | (0.095) | (0.665) | (0.087) |
| Size | 0.002*** | 0.003*** | 0.002*** | 0.003*** | 0.001 | 0.003 | 0.001 | 0.003 |
| | (0.005) | (0.005) | (0.004) | (0.000) | (0.898) | (0.627) | (0.875) | (0.611) |
| GDP Growth | -0.230*** | -0.217*** | -0.272*** | -0.264*** | -0.181*** | -0.170*** | -0.203*** | -0.204*** |
| | (0.000) | (0.000) | (0.000) | (0.000) | (0.000) | (0.001) | (0.000) | (0.000) |
| Inflation | 0.014 | 0.010 | 0.012 | 0.004 | 0.048 | 0.047 | 0.047 | 0.039 |
| | (0.682) | (0.761) | (0.729) | (0.899) | (0.136) | (0.137) | (0.145) | (0.215) |
| ISLAMIC_D | -0.004 | -0.001 | -0.004 | -0.001 | -0.001 | 0.001 | 0.001 | 0.001 |
| | (0.330) | (0.730) | (0.343) | (0.748) | (0.996) | (0.691) | (0.990) | (0.676) |
| CRISIS_D | -0.038*** | -0.038*** | -0.042*** | -0.042*** | -0.016* | -0.017* | -0.018* | -0.020* |
| | (0.002) | (0.001) | (0.001) | (0.000) | (0.108) | (0.106) | (0.099) | (0.056) |
| Country Dummy | Yes | Yes | Yes | Yes | Yes | Yes | Yes | Yes |
| Year Dummy | Yes | Yes | Yes | Yes | Yes | Yes | Yes | Yes |
| Number of Observations | 2901 | 2901 | 2901 | 2901 | 2901 | 2901 | 2901 | 2901 |
| R-squared (Overall) | 0.2713 | 0.2978 | 0.2714 | 0.2974 | 0.2733 | 0.2919 | 0.2718 | 0.2912 |

The panel data regressions estimate the relation between banking competition and capital ratios over the period of 2006–2018 while controlling for important bank-level and macroeconomic characteristics. The sample includes 225 banks in 18 countries in the MENA region. Banks included in the sample are only Islamic banks. As a measure of bank capitalization level we use the Total eligible capital to total assets (EC/TA) and Total equity to total liability (TE/TL) ratios. All the regressions control for year and country fixed effects.

*, **, and *** indicate statistical significance at the 10%, 5%, and 1% level, respectively. The values in parenthesis represent 'p-value'. Capital adequacy ratio, market competition indicators, bank-level characteristics, institution, ownership and macroeconomic variables are described in S1 Appendix.

of financial stability. This finding does not coincide with the predictions of the Structure-Conduct-Performance (SCP) hypothesis, which postulates that highly concentrated firms are more competitive and profitable, and have more market power that allows them to achieve higher financial stability. Under the SCP paradigm, concentration measures such as HHI are used as proxies for competition [73, 84]. Therefore, the SCP paradigm assumes that banks operating in concentrated markets have a higher profitability due to monopoly rents.

Individually, the coefficient of H-Statistic in Table 6 is statistically significant and negative (see Model 3) which contradicts the findings of previous research on the MENA region as reported by [1]. Further, the estimated coefficient of the interaction term of both measures of competition in Model 4 is negative yet insignificant; therefore, we cannot confirm the notion that increased concentration does have to be associated with uncompetitive markets. If this is the case, regulatory authorities and decision makers in the MENA region may force more concentrated banks to avoid risky strategies through appropriate actions that increase banking competition.

The relationship between the index of activity restriction and credit risk is positive and significant; therefore, regulators may influence banks to keep their credit risk low through stringent restrictions on their activities in order to increase the overall stability of the banking system. We also find that the impact on credit risk decreases for banks with higher level of ownership concentration and is less pronounced in countries with strong institutions. The results reported in Model 5 to 8 provide further support to our findings. The most significant difference relates to the moderating effect of market competition (measured by the H-Statistic) on the relationship between market concentration and bank risk. We observe that the negative impact of market concentration on bank financial stability is reduced when banking competition is increasing. Regarding the bank-level and macroeconomic control variables, bank size shows a positive (negative) effect on risk level (financial stability), thus supporting the 'moral hazard" hypothesis and the 'too big to fail' proposition that the larger the bank size, the greater the chance of raising risk and lessening financial stability. From a macroeconomic point of view, both the GDP growth and inflation show a significant impact on risk. While higher GDP growth warrants better banks' financial stability higher inflation would discount the financial soundness of MENA banks.

## The differential effect on Islamic banks' risk behavior

The question of whether the impact of competition and capital ratios on bank stability is significantly different between CBs and IBs remains unexplored so far. To the best of our knowledge we are the first to have addressed this important question with strong policy implications. Therefore, we run our analysis separately for the samples of CBs and IBs. The outputs of the regression analysis are reported in Tables 7 and 8 (**see S2 and S3 Files**).

In Table 7, we report the results for CBs sample using LLR/GL ratio as a measure of bank credit risk. In line with our first hypothesis (H1b) according to which capital and competition have a significant impact on bank risk, we find that capital ratio exerts a strong positive influence on risk behavior of CBs (see Model 1). As evident from previous research [7, 10], less activity restrictions are associated with improved banking stability. We provide further support to this general notion as the relationship between the index of activity restriction and credit risk is positive and strongly significant. Next, we observe that the HH-index enters the regressions in Table 7 positively and significantly, indicating that more concentrated banks usually keep higher levels of credit risk. In opposite, the coefficient of H-Statistic is statistically significant and negative, which supports the general notion that concentration and competition are inversely related. Our results complement the findings of [1] who report a similar

**Table 7. Panel regressions of competition and credit risk (CBs, 2006–2018).**

| Explanatory Variables | LLR/GL | | | | NPL/GL | | | |
|---|---|---|---|---|---|---|---|---|
| | Model 1 | Model 2 | Model 3 | Model 4 | Model 5 | Model 6 | Model 7 | Model 8 |
| Constant | 0.025 | -0.031 | 0.037 | -0.018 | 0.028 | -0.009 | 0.037 | 0.001 |
| | (0.608) | (0.530) | (0.455) | (0.708) | (0.488) | (0.822) | (0.361) | (0.980) |
| EC/TA | 0.158*** | 0.140*** | 0.159*** | 0.140*** | 0.134*** | 0.117*** | 0.134*** | 0.116*** |
| | (0.000) | (0.000) | (0.000) | (0.000) | (0.000) | (0.000) | (0.000) | (0.000) |
| HHI | | 0.138*** | | 0.138*** | | 0.091*** | | 0.090*** |
| | | (0.000) | | (0.000) | | (0.000) | | (0.000) |
| H-Stat | | | -0.023** | -0.022** | | | -0.017** | -0.016** |
| | | | (0.020) | (0.024) | | | (0.033) | (0.046) |
| | | | | -0.005 | | | | -0.010** |
| HHI*H-Stat | | | | (0.341) | | | | (0.022) |
| Activity index | 0.001* | 0.001** | 0.001** | 0.001*** | 0.001*** | 0.001*** | 0.001*** | 0.001*** |
| | (0.058) | (0.022) | (0.026) | (0.008) | (0.009) | (0.005) | (0.004) | (0.001) |
| Ownership concentration | -0.017*** | -0.016*** | -0.017*** | -0.016** | -0.013*** | -0.013*** | -0.013*** | -0.013*** |
| | (0.006) | (0.008) | (0.005) | (0.008) | (0.006) | (0.009) | (0.006) | (0.009) |
| Institution | 0.007 | 0.008 | 0.006 | 0.006 | -0.003 | -0.003 | -0.004 | -0.004 |
| | (0.255) | (0.219) | (0.357) | (0.308) | (0.508) | (0.561) | (0.392) | (0.448) |
| Cost-income ratio | 0.006*** | 0.006*** | 0.006*** | 0.006*** | 0.001*** | 0.001*** | 0.001*** | 0.001*** |
| | (0.000) | (0.000) | (0.000) | (0.000) | (0.000) | (0.000) | (0.000) | (0.000) |
| Deposit/Total Assets | 0.033*** | 0.023*** | 0.033*** | 0.023*** | 0.080*** | 0.068*** | 0.080*** | 0.068*** |
| | (0.000) | (0.007) | (0.000) | (0.006) | (0.000) | (0.000) | (0.000) | (0.000) |
| Loan/ Total Assets | 0.052*** | 0.011* | 0.050** | 0.010* | -0.095*** | -0.086*** | -0.096*** | -0.086*** |
| | (0.000) | (0.053) | (0.001) | (0.502) | (0.000) | (0.000) | (0.000) | (0.000) |
| Loan/Total Deposit | -0.086*** | -0.074*** | -0.084*** | -0.073*** | -0.040 | -0.045*** | -0.039*** | -0.044*** |
| | (0.000) | (0.000) | (0.000) | (0.000) | (0.000) | (0.000) | (0.000) | (0.000) |
| Revenue Diversification | 1.868*** | 1.691*** | 1.811*** | 1.643*** | 1.407*** | 1.391*** | 1.371*** | 1.364*** |
| | (0.000) | (0.000) | (0.000) | (0.000) | (0.000) | (0.000) | (0.000) | (0.000) |
| Pre-tax ROA | -0.634*** | -0.736*** | -0.599*** | -0.705*** | -0.286** | -0.435*** | -0.265** | -0.418*** |
| | (0.000) | (0.000) | (0.000) | (0.000) | (0.022) | (0.001) | (0.034) | (0.001) |
| Size | 0.003*** | 0.003*** | 0.003*** | 0.003*** | 0.002*** | 0.002*** | 0.002*** | 0.002*** |
| | (0.000) | (0.000) | (0.000) | (0.000) | (0.003) | (0.001) | (0.004) | (0.002) |
| GDP Growth | -0.312*** | -0.302*** | -0.335*** | -0.330*** | -0.214* | -0.202*** | -0.231*** | -0.230*** |
| | (0.000) | (0.000) | (0.000) | (0.000) | (0.081) | (0.000) | (0.000) | (0.000) |
| Inflation | 0.021 | 0.026 | 0.023 | 0.024 | -0.011 | 0.068* | 0.064* | 0.061* |
| | (0.633) | (0.540) | (0.599) | (0.579) | (0.282) | (0.054) | (0.073) | (0.085) |
| CRISIS_D | -0.029** | -0.027** | -0.031** | -0.030** | -0.011 | -0.011 | -0.013 | -0.014 |
| | (0.025) | (0.038) | (0.017) | (0.022) | (0.282) | (0.302) | (0.223) | (0.188) |
| Country Dummy | Yes | Yes | Yes | Yes | Yes | Yes | Yes | Yes |
| Year Dummy | Yes | Yes | Yes | Yes | Yes | Yes | Yes | Yes |
| Number of Observations | 2106 | 2106 | 2106 | 2106 | 2106 | 2106 | 2106 | 2106 |
| R-squared (Overall) | 0.3884 | 0.4036 | 0.3874 | 0.4025 | 0.4242 | 0.4398 | 0.4233 | 0.4394 |

The panel data regressions estimate the relation between banking competition and credit risk over the period of 2006–2018 while controlling for important bank-level and macroeconomic characteristics. The sample includes 225 banks in 18 countries in the MENA region. Banks included in the sample are only conventional banks. As a measure of bank credit risk we use Loan loss reserves to gross loans (LLR/GL) and Non-performing loans to gross loans (NPL/GL) ratios. All the regressions control for year and country fixed effects.

*, **, and *** indicate statistical significance at the 10%, 5%, and 1% level, respectively. The values in parenthesis represent 'p-value'. Capital adequacy ratio, market competition indicators, bank-level characteristics, institution, ownership and macroeconomic variables are described in S1 Appendix.

**Table 8. Panel regressions of competition and credit risk (IBs, 2006–2018).**

| Explanatory Variables | LLR/GL | | | | NPL/GL | | | |
|---|---|---|---|---|---|---|---|---|
| | Model 1 | Model 2 | Model 3 | Model 4 | Model 5 | Model 6 | Model 7 | Model 8 |
| Constant | -0.060 | -0.047 | -0.052 | -0.040 | -0.086 | -0.075 | -0.088 | -0.077 |
| | (0.291) | (0.404) | (0.364) | (0.479) | (0.159) | (0.213) | (0.152) | (0.206) |
| EC/TA | 0.024*** | 0.021*** | 0.023*** | 0.020*** | 0.039*** | 0.036*** | 0.040*** | 0.036*** |
| | (0.000) | (0.001) | (0.000) | (0.000) | (0.000) | (0.000) | (0.000) | (0.000) |
| HHI | | 0.154*** | | 0.152*** | | 0.120*** | | 0.121*** |
| | | (0.000) | | (0.000) | | (0.001) | | (0.001) |
| H-Stat | | | -0.024* | -0.020 | | | 0.005 | 0.006 |
| | | | (0.098) | (0.157) | | | (0.744) | (0.690) |
| | | | | -0.002 | | | | -0.005 |
| HHI*H-Stat | | | | (0.757) | | | | (0.544) |
| Activity index | 0.009 | 0.007 | 0.001 | 0.001 | -0.001 | -0.001 | -0.001 | -0.001 |
| | (0.442) | (0.513) | (0.338) | (0.398) | (0.336) | (0.278) | (0.320) | (0.286) |
| Ownership concentration | 0.008 | 0.005 | 0.003 | 0.004 | 0.004 | 0.007 | 0.006 | 0.001 |
| | (0.946) | (0.661) | (0.976) | (0.726) | (0.976) | (0.958) | (0.963) | (0.938) |
| Institution | -0.049*** | -0.046*** | -0.047*** | -0.043*** | -0.026** | -0.024* | -0.027** | -0.024* |
| | (0.000) | (0.000) | (0.000) | (0.000) | (0.041) | (0.061) | (0.039) | (0.058) |
| Cost-income ratio | 0.007* | 0.008** | 0.007** | 0.008** | 0.008** | 0.009** | 0.008** | 0.010** |
| | (0.054) | (0.029) | (0.048) | (0.027) | (0.041) | (0.019) | (0.041) | (0.018) |
| Deposit/Total Assets | 0.014 | 0.005 | 0.013 | 0.004 | 0.040** | 0.027* | 0.040** | 0.027* |
| | (0.285) | (0.679) | (0.324) | (0.731) | (0.016) | (0.109) | (0.015) | (0.109) |
| Loan/ Total Assets | 0.063*** | 0.033** | 0.066*** | 0.037*** | -0.017 | -0.015 | -0.017 | -0.015 |
| | (0.005) | (0.043) | (0.003) | (0.008) | (0.442) | (0.474) | (0.439) | (0.475) |
| Loan/Total Deposit | -0.051*** | -0.059*** | -0.051*** | -0.059*** | -0.071*** | -0.088*** | -0.072*** | -0.089*** |
| | (0.000) | (0.000) | (0.000) | (0.000) | (0.000) | (0.000) | (0.000) | (0.000) |
| Revenue Diversification | 0.258*** | 0.246*** | 0.270*** | 0.257*** | 0.328*** | 0.313*** | 0.326*** | 0.310*** |
| | (0.000) | (0.000) | (0.000) | (0.000) | (0.000) | (0.000) | (0.000) | (0.000) |
| Pre-tax ROA | -0.248* | -0.291** | -0.257** | -0.299** | -0.276** | -0.328** | -0.276** | -0.329** |
| | (0.050) | (0.020) | (0.042) | (0.017) | (0.039) | (0.014) | (0.039) | (0.014) |
| Size | 0.002 | -0.002* | 0.002 | -0.002* | -0.006*** | -0.006*** | -0.007*** | -0.006*** |
| | (0.171) | (0.061) | (0.151) | (0.054) | (0.001) | (0.003) | (0.001) | (0.003) |
| GDP Growth | -0.180** | -0.183** | -0.212** | -0.213** | 0.001 | -0.046 | -0.047 | -0.045 |
| | (0.045) | (0.038) | (0.021) | (0.019) | (0.632) | (0.681) | (0.686) | (0.699) |
| Inflation | 0.069 | 0.058 | 0.057 | 0.047 | -0.034 | -0.035 | -0.032 | -0.034 |
| | (0.186) | (0.257) | (0.274) | (0.360) | (0.602) | (0.591) | (0.623) | (0.599) |
| CRISIS_D | -0.030*** | -0.036*** | -0.032*** | -0.037*** | -0.010* | -0.020** | -0.009* | -0.020** |
| | (0.006) | (0.001) | (0.003) | (0.001) | (0.088) | (0.036) | (0.105) | (0.023) |
| Country Dummy | Yes | Yes | Yes | Yes | Yes | Yes | Yes | Yes |
| Year Dummy | Yes | Yes | Yes | Yes | Yes | Yes | Yes | Yes |
| Number of Observations | 795 | 795 | 795 | 795 | 795 | 795 | 795 | 795 |
| R-squared (Overall) | 0.2081 | 0.2296 | 0.2110 | 0.2317 | 0.1458 | 0.1629 | 0.1459 | 0.1641 |

The panel data regressions estimate the relation between banking competition and credit risk over the period of 2006–2018 while controlling for important bank-level and macroeconomic characteristics. The sample includes 225 banks in 18 countries in the MENA region. Banks included in the sample are only Islamic banks. As a measure of bank credit risk we use Loan loss reserves to gross loans (LLR/GL) and Non-performing loans to gross loans (NPL/GL) ratios. All the regressions control for year and country fixed effects.

*, **, and *** indicate statistical significance at the 10%, 5%, and 1% level, respectively The values in parenthesis represent 'p-value'. Capital adequacy ratio, market competition indicators, bank-level characteristics, institution, ownership and macroeconomic variables are described in S1 Appendix.

negative relationship between bank risk and market competition for the group of Gulf countries, and show that an increase in competition in countries with an average moderate level leads to an improvement in the financial stability.

The results in Table 8 lend some thoughts for future discussions of the competition effect on IBs. For example, in both samples the association between concentration and bank credit risk is strongly positive, of almost the same magnitude. Thus, our second hypotheses (H2a) is to be rejected. However, the effect of market competition (measured by the H-Statistic) is negative yet insignificant in the sample of IBs. This finding suggests that increased competition in the banking market has no impact on the credit risk decisions of IBs. On the other side, the positive association of concentration with the risk-taking behavior of IBs and CBs provides further support to our second hypothesis (H2b), which postulates that the impact of competition on bank risk is expected to be similar between Islamic and conventional banking. Second, we are unable to provide support to the notion that increased competition may impact on the relationship between the level of concentration and risk-taking behavior of IBs. Third, we observe a strongly significant and positive relationship between the index of activity restriction and bank risk in the group of CBs but this effect seems to be insignificant for IBs.

Finally, the negative coefficient of the crisis dummy variable for IBs indicates that credit risk levels are lower in the crisis period than in the non-crisis periods. This finding is further supported by the recent research on COVID-19 pandemic. For example [29], finds that IBs have lower incentive to undertake higher credit risk than conventional banking institutions, which is also confirmed by previous results of [99], and [32]. This behavior of IBs is even more pronounced during the COVID-19 pandemic and shows their superiority to other types of banks due to their less risk aptitude and financial stability. Again, our findings are important for regulators and policy makers in the MENA countries as they inform regulatory authorities for the need to set the level of regulations (capital requirements and activity restrictions) in such a way that prevents concentrated banks from engaging in risky activities in the face of increased competition. These restrictions will have more significant effect on Islamic banking market which is characterized with higher level of banking concentration (see Table 2).

## Robustness checks and alternative specifications

We perform a number of robustness tests. First, in addition to the fixed and random effect models reported in previous tables, the analysis employs identical specifications (see Eq (1) and (2)) using the Generalized Method of Moments (GMM) estimator, developed by [100]. This estimator controls for the presence of unobserved firm-specific effects and for the endogeneity of explanatory variables. The instruments used depend on the assumption made as to whether the variables are endogenous or predetermined, or exogenous. The validity of the instruments is tested using a Sargan test of over-identifying restrictions and a test of the absence of serial correlation of the residuals. The AR(2) test detects the second-order autocorrelation in first differences. We treat the lagged dependent variables as endogenous, so that GMM-style instruments of deeper lags are created. The results of the GMM tests for market competition effects are reported in Tables 9 (**see S1 File**) and 10 (**see S1 File**). Specifically, in Table 9, we present the individual effects of credit risk and market competition on bank capitalization level, whereas in the last model we estimate the interaction effect between concentration and completion measured by the HH-index and the H-Statistic, respectively. We run similar regressions for bank credit risk in Table 10. The results support our findings that banks raise their capitalization levels in response to a higher risk but the opposite is also true. Moreover, market competition seems to play an important role in explaining the capitalization level and risk-taking behavior of banks in the MENA region.

**Table 9. Panel regressions (GMM) of competition and capital ratio (all banks, 2006–2018).**

| Explanatory Variables | EC/TA | | | | TE/TA | | | |
|---|---|---|---|---|---|---|---|---|
| | Model 1 | Model 2 | Model 3 | Model 4 | Model 5 | Model 6 | Model 7 | Model 8 |
| Constant | -0.011 | -0.024 | 0.008 | 0.001 | 0.015 | 0.021 | 0.024 | 0.034 |
| | (0.900) | (0.801) | (0.924) | (0.982) | (0.753) | (0.662) | (0.648) | (0.510) |
| Credit risk | 0.591*** | 0.562*** | 0.577*** | 0.542*** | 0.001* | 0.012* | 0.002* | 0.010** |
| | (0.000) | (0.000) | (0.000) | (0.000) | (0.073) | (0.106) | (0.059) | (0.028) |
| HHI | | 0.110* | | 0.108* | | -0.040*** | | -0.040*** |
| | | (0.066) | | (0.055) | | (0.002) | | (0.001) |
| H-Stat | | | -0.073 | -0.073* | | | 0.005 | 0.006** |
| | | | (0.110) | (0.104) | | | (0.759) | (0.017) |
| | | | | -0.035** | | | | -0.004 |
| HHI*H-Stat | | | | (0.023) | | | | (0.818) |
| Activity index | -0.002 | -0.002 | -0.001 | -0.001 | 0.001 | 0.001 | 0.001 | 0.001 |
| | (0.183) | (0.257) | (0.325) | (0.506) | (0.282) | (0.298) | (0.299) | (0.303) |
| Ownership concentration | -0.011 | -0.011 | -0.011 | -0.011 | -0.005 | -0.007 | -0.005 | -0.007 |
| | (0.581) | (0.604) | (0.581) | (0.588) | (0.593) | (0.491) | (0.595) | (0.478) |
| Institution | -0.006* | -0.008* | -0.003* | -0.007* | -0.012* | -0.013* | -0.012* | -0.013* |
| | (0.057) | (0.085) | (0.049) | (0.024) | (0.098) | (0.055) | (0.075) | (0.036) |
| Cost-income ratio | -0.007*** | -0.007*** | -0.007*** | -0.007*** | -0.001* | -0.001* | -0.001* | -0.001* |
| | (0.001) | (0.003) | (0.003) | (0.008) | (0.085) | (0.025) | (0.102) | (0.054) |
| Deposit/Total Assets | -0.146*** | -0.152*** | -0.145*** | -0.151*** | 0.018 | 0.020 | 0.019 | 0.020 |
| | (0.003) | (0.005) | (0.003) | (0.004) | (0.402) | (0.366) | (0.395) | (0.362) |
| Loan/ Total Assets | 0.295*** | 0.272*** | 0.299*** | 0.279*** | 0.025** | 0.025* | 0.025* | 0.024* |
| | (0.000) | (0.000) | (0.000) | (0.000) | (0.047) | (0.059) | (0.054) | (0.054) |
| Loan/Total Deposit | -0.051 | -0.051 | -0.055 | -0.056 | -0.003 | -0.001 | -0.003 | -0.001 |
| | (0.394) | (0.395) | (0.364) | (0.351) | (0.555) | (0.862) | (0.575) | (0.872) |
| Revenue Diversification | 0.536 | 0.535 | 0.554 | 0.568 | 0.238** | 0.253** | 0.239** | 0.259** |
| | (0.222) | (0.225) | (0.212) | (0.208) | (0.045) | (0.029) | (0.042) | (0.024) |
| Leverage | 0.137*** | 0.135*** | 0.140*** | 0.137*** | 0.012** | 0.012* | 0.012*** | 0.012** |
| | (0.000) | (0.000) | (0.000) | (0.000) | (0.032) | (0.058) | (0.009) | (0.031) |
| Pre-tax ROA | -1.754*** | -1.832*** | -1.711*** | -1.812*** | -0.168** | -0.114* | -0.174** | -0.125* |
| | (0.000) | (0.000) | (0.000) | (0.000) | (0.012) | (0.072) | (0.016) | (0.050) |
| Size | 0.004 | 0.004 | 0.004 | 0.004 | 0.003* | 0.003* | 0.003* | 0.003* |
| | (0.264) | (0.248) | (0.275) | (0.266) | (0.092) | (0.106) | (0.090) | (0.087) |
| GDP Growth | 0.395*** | 0.397*** | 0.313*** | 0.268** | -0.215** | -0.211** | -0.225*** | -0.237*** |
| | (0.002) | (0.003) | (0.000) | (0.012) | (0.016) | (0.020) | (0.004) | (0.006) |
| Inflation | 0.017 | 0.013 | 0.012 | -0.006 | -0.157*** | -0.160*** | -0.156*** | -0.163*** |
| | (0.914) | (0.935) | (0.937) | (0.966) | (0.000) | (0.000) | (0.000) | (0.000) |
| Lag_1(Dependent) | -0.009 | -0.010 | -0.009 | -0.010 | -0.001 | -0.005 | -0.002 | -0.005 |
| | (0.181) | (0.277) | (0.189) | (0.274) | (0.953) | (0.869) | (0.936) | (0.850) |
| ISLAMIC_D | 0.124*** | 0.126*** | 0.123*** | 0.127*** | 0.065*** | 0.064*** | 0.065*** | 0.065*** |
| | (0.000) | (0.000) | (0.000) | (0.000) | (0.000) | (0.000) | (0.000) | (0.000) |
| CRISIS_D | -0.005 | -0.007 | -0.017 | -0.023 | -0.005 | -0.004 | -0.004 | 0.002 |
| | (0.711) | (0.608) | (0.374) | (0.233) | (0.483) | (0.566) | (0.634) | (0.755) |
| Country Dummy | Yes | Yes | Yes | Yes | Yes | Yes | Yes | Yes |
| Chi Sq. | 1715.08*** | 1475.55*** | 1052.44*** | 843.80*** | 104.66*** | 85.42** | 139.07*** | 91.64** |
| Number of observations | 2827 | 2827 | 2827 | 2827 | 2827 | 2827 | 2827 | 2827 |
| Number of instruments | 481 | 484 | 484 | 490 | 505 | 508 | 508 | 514 |

(*Continued*)

**Table 9.** (Continued)

| Explanatory Variables | EC/TA | | | | TE/TA | | | |
|---|---|---|---|---|---|---|---|---|
| | Model 1 | Model 2 | Model 3 | Model 4 | Model 5 | Model 6 | Model 7 | Model 8 |
| P-value for AR(2) tests | 0.6385 | 0.6078 | 0.8300 | 0.8042 | 0.9631 | 0.9556 | 0.9835 | 0.9187 |

The panel data regressions estimate the relation between banking competition and capital ratios over the period of 2006–2018 while controlling for important bank-level and macroeconomic characteristics. The sample includes 225 banks in 18 countries in the MENA region. Banks included in the sample are conventional banks (162) and Islamic banks (63). As a measure of bank capitalization level we use the Total eligible capital to total assets (EC/TA) and Total equity to total liability (TE/TL) ratios. All the regressions control for year and country fixed effects.

*, **, and *** indicate statistical significance at the 10%, 5%, and 1% level, respectively. The values in parenthesis represent 'p-value'. Capital adequacy ratio, market competition indicators, bank-level characteristics, institution, ownership and macroeconomic variables are described in S1 Appendix. We use Arellano–Bond test for serial correlation in the first-differenced errors at order 'm'. We reject the null hypothesis of no serial correlation in the first-differenced errors but accept the null hypothesis of no serial correlation in the second-differenced errors. In addition, we use the first two lags of all independent variables as additional instruments.

Second, we investigate the robustness of our results using alternative specifications and different control variables. For example, in addition to distance to default and equity volatility, we proxy bank insolvency risk with credit default swap (CDS) spread. The results are inconclusive due to the limited number of available observations for CDS. We follow [2] in using changes in deposits and loans to proxy for deposit and credit risk. The estimation results show that the relationship between capital requirements and deposit changes is significant in both samples, whereas, the level of market competition has a strong influence on bank risk behavior only in the sample of CBs (the results are not reported here but are available on request). Next, we include an alternative measure of the degree of competition (Lerner indicator as a measure of market power). The regression outputs show positive and significant impact of market power on risk behavior of banks in the MENA region. Finally, we split the sample into two sub-periods, before the global financial crisis of 2008–2009 and after the crisis. In line with [1] we find no differences in the effect of market competition on bank risk in terms of whether the economy is in an expansive or recessive moment.

## Conclusions

This paper investigates the influence of market competition and risk on capitalization level of banks in the MENA region. Most of the previous studies on the MENA region examine the impact of market competition on credit risk only [2] or the banking system in the MENA region as a whole [1]. The differential impact on IBs is also not well documented. We extend the existing empirical literature by providing new evidence on the impact of competition and risk on capital levels of banks that has strong implications for the banking system performance during the COIVD-19 pandemic.

An early study by [30] on the MENA region finds that banks raise their capitalization levels in response to a higher risk rather than the other way round. We test this hypothesis using a larger sample of banks (162 CBs and 63 IBs) in the MENA region, and find a positive association between bank capital and risk, which provides further support to the regulatory hypothesis [69]. Our evidence confirms that banks do increase their capitalization levels in response to a higher risk. However, the opposite is also true–banks are taking more risk in a response to increased capital requirements set by the regulatory authorities. The analysis of the impact of competition on banks' capital level allows to have addressed a number of important questions with strong policy implications. First, the increased level of banking concentration forces the banks in the MENA region to increase correspondingly their capitalization levels. Second, the level of market competition (measured by the H-Statistic) has no significant impact on a

**Table 10. Panel regressions (GMM) of competition and credit risk (all banks, 2006–2018).**

| Explanatory Variables | LLR/GL | | | | NPL/GL | | | |
|---|---|---|---|---|---|---|---|---|
| | Model 1 | Model 2 | Model 3 | Model 4 | Model 5 | Model 6 | Model 7 | Model 8 |
| Constant | -0.008 | -0.0161 | 0.009 | -0.004 | -0.019 | -0.036 | 0.001 | -0.015 |
| | (0.828) | (0.663) | (0.816) | (0.991) | (0.512) | (0.176) | (0.962) | (0.579) |
| EC/TA | 0.037* | 0.029* | 0.036* | 0.028* | 0.071* | 0.064* | 0.071* | 0.064* |
| | (0.097) | (0.104) | (0.100) | (0.101) | (0.054) | (0.071) | (0.055) | (0.071) |
| HHI | | 0.170*** | | 0.168*** | | 0.119*** | | 0.116*** |
| | | (0.000) | | (0.000) | | (0.000) | | (0.000) |
| H-Stat | | | -0.034* | -0.029* | | | -0.021* | -0.019* |
| | | | (0.097) | (0.106) | | | (0.063) | (0.074) |
| HHI*H-Stat | | | | -0.006** | | | | -0.012* |
| | | | | (0.012) | | | | (0.106) |
| Activity index | 0.001* | 0.001** | 0.001* | 0.002*** | 0.003* | 0.005* | 0.005* | 0.008* |
| | (0.100) | (0.028) | (0.090) | (0.004) | (0.056) | (0.065) | (0.086) | (0.100) |
| Ownership concentration | -0.009* | -0.007* | -0.009* | -0.008* | -0.003 | 0.008 | -0.002 | 0.006 |
| | (0.089) | (0.100) | (0.080) | (0.107) | (0.947) | (0.869) | (0.959) | (0.891) |
| Institution | -0.013 | -0.010 | -0.015 | -0.011 | -0.006 | -0.003 | -0.007 | -0.004 |
| | (0.525) | (0.594) | (0.477) | (0.553) | (0.734) | (0.836) | (0.698) | (0.823) |
| Cost-income ratio | 0.008** | 0.008** | 0.008** | 0.008** | 0.001*** | 0.001*** | 0.001*** | 0.001*** |
| | (0.040) | (0.040) | (0.036) | (0.037) | (0.000) | (0.000) | (0.000) | (0.000) |
| Deposit/Total Assets | 0.020** | 0.009* | 0.019** | 0.009* | 0.032* | 0.015* | 0.032* | 0.016* |
| | (0.014) | (0.059) | (0.015) | (0.054) | (0.088) | (0.059) | (0.089) | (0.017) |
| Loan/ Total Assets | 0.037* | -0.001 | 0.040* | 0.002 | -0.043* | -0.028 | -0.044* | -0.030 |
| | (0.071) | (0.920) | (0.061) | (0.907) | (0.075) | (0.252) | (0.073) | (0.228) |
| Loan/Total Deposit | -0.036*** | -0.034*** | -0.037*** | -0.035*** | -0.023*** | -0.032*** | -0.023*** | -0.031*** |
| | (0.000) | (0.000) | (0.000) | (0.000) | (0.005) | (0.000) | (0.007) | (0.000) |
| Revenue Diversification | 0.279** | 0.262** | 0.293** | 0.276** | 0.253** | 0.261** | 0.262** | 0.274*** |
| | (0.024) | (0.028) | (0.025) | (0.028) | (0.012) | (0.011) | (0.010) | (0.009) |
| Pre-tax ROA | 0.088 | -0.015 | 0.075 | -0.022 | 0.124 | -0.020 | -0.108 | -0.034 |
| | (0.574) | (0.920) | (0.648) | (0.883) | (0.601) | (0.932) | (0.654) | (0.886) |
| Size | 0.004*** | 0.005*** | 0.004*** | 0.005*** | 0.002 | 0.002* | 0.002* | 0.002* |
| | (0.000) | (0.000) | (0.000) | (0.000) | (0.134) | (0.066) | (0.101) | (0.059) |
| GDP Growth | -0.356*** | -0.338*** | -0.398*** | -0.383*** | -0.307*** | -0.289*** | -0.342*** | -0.335*** |
| | (0.000) | (0.000) | (0.000) | (0.000) | (0.000) | (0.000) | (0.000) | (0.000) |
| Inflation | 0.090* | 0.084* | 0.093* | 0.084* | 0.074 | 0.072 | 0.078 | 0.071 |
| | (0.089) | (0.109) | (0.096) | (0.103) | (0.156) | (0.185) | (0.138) | (0.205) |
| Lag_1(Dependent) | -0.011 | -0.010 | -0.008 | -0.008 | 0.034 | 0.041 | 0.031 | 0.037 |
| | (0.593) | (0.628) | (0.705) | (0.686) | (0.246) | (0.186) | (0.274) | (0.219) |
| ISLAMIC_D | -0.015** | -0.013** | -0.016** | -0.013** | -0.006 | -0.004 | -0.006 | -0.003 |
| | (0.019) | (0.028) | (0.016) | (0.019) | (0.199) | (0.432) | (0.214) | (0.451) |
| CRISIS_D | -0.099*** | -0.096*** | -0.103*** | -0.100*** | -0.071*** | -0.069*** | -0.075*** | -0.074*** |
| | (0.001) | (0.000) | (0.001) | (0.000) | (0.008) | (0.004) | (0.007) | (0.004) |
| Country Dummy | Yes | Yes | Yes | Yes | Yes | Yes | Yes | Yes |
| Chi Sq. | 379.42*** | 1653.17*** | 893.18*** | 1281.44*** | 156.2*** | 2552.82*** | 3134.36*** | 556.24.03** |
| Number of observations | 2827 | 2827 | 2827 | 2827 | 2827 | 2827 | 2827 | 2827 |
| Number of instruments | 484 | 487 | 487 | 493 | 490 | 493 | 493 | 499 |

(*Continued*)

**Table 10.** (Continued)

| Explanatory Variables | LLR/GL | | | | NPL/GL | | | |
|---|---|---|---|---|---|---|---|---|
| | Model 1 | Model 2 | Model 3 | Model 4 | Model 5 | Model 6 | Model 7 | Model 8 |
| P-value for AR(2) tests | 0.2732 | 0.5431 | 0.2742 | 0.5797 | 0.8631 | 0.5717 | 0.8138 | 0.5998 |

The panel data regressions estimate the relation between banking competition and credit risk over the period of 2006–2018 while controlling for important bank-level and macroeconomic characteristics. The sample includes 225 banks in 18 countries in the MENA region. Banks included in the sample are conventional banks (162) and Islamic banks (63). As a measure of bank credit risk we use Loan loss reserves to gross loans (LLR/GL) and Non-performing loans to gross loans (NPL/GL) ratios. All the regressions control for year and country fixed effects.

*, **, and *** indicate statistical significance at the 10%, 5%, and 1% level, respectively. Bank-level characteristics, market competition, risk measure, ownership and macroeconomic variables are described in S1 Appendix. We use Arellano–Bond test for serial correlation in the first-differenced errors at order 'm'. We reject the null hypothesis of no serial correlation in the first-differenced errors but accept the null hypothesis of no serial correlation in the second-differenced errors. In addition, we use the first two lags of all independent variables as additional instruments.

bank's decision to increase or decrease its capitalization level. Instead, it will depend on the regulatory authority's behavior and not on the level of banking competition in the country. Finally, we are able to confirm the notion that increased concentration does not have to be associated with uncompetitive markets. We find a positive relation between concentration and completion when total equity to total liability ratio is used as a proxy for credit risk.

Previous research does not provide a consistent answer to the question of whether the impact of competition and credit risk on bank capital ratios is significantly different between CBs and IBs. To the best of our knowledge this is the only study to have addressed this important question with strong policy implications. We find a significant differential effect of competition on capital ratios of IBs. Specifically, we observe a positive association between banking concentration and capital for either bank; however, the effect of market competition is strongly significant and negative only in the sample of IBs. The negative association between competition and bank capital indicates that an increase in banking competition in the MENA countries will lead to a reduction in the financial soundness of IBs. Thus, our finding disagrees with [2, 74] who obtained a positive relationship between competition and concentration. Furthermore, we find strong evidence for the moderating role of marker competition on the link between concentration and bank capitalization only in the sample of CBs, while this effect is insignificant for IBs.

We also provide evidence that banks raise their risk levels in response to increased concentration. More specifically, the positive linear relationship between the HH-index and credit risk indicates that an increase in banking concentration may reduce financial stability of the banking system as a whole. In opposite, the negative coefficient of H-Statistic speaks for an inverse association between market competition and bank risk, which contradicts the findings of previous research on the MENA region that reports a positive link [1]. Our results also disagree with the predictions of the Structure-Conduct-Performance (SCP) hypothesis, which postulates that higher concentration would lead to less competition and, consequently, greater financial stability. We contribute to the empirical literature that deals with competition effect by exploring the moderating effect of market competition on the relationship between bank risk and the level of concentration. We find that more concentrated banks increase their risk level when they have higher marker power in lending but this decision is not dictated by the level of competition in the banking marker. This result calls into question the strategy taken by decision makers in some MENA countries to increase the level of banking concentration in order to improve financial stability.

Our results have strong implications for regulators, policy makers and bank managers in the context of the current pandemic situation. First, banks in MENA region seem to raise their capitalization levels in response to a higher risk; however, the capital requirements increase it. Further, market competition and concentration also play an important role in risk-taking behavior of banks. Therefore, regulators and policy makers in the MENA region should introduce policies that restrict the risky behavior of banks through stringent capital requirements and more intense banking supervision to prevent them from taking excessive risk. Second, our findings suggest that, in highly concentrated market, Islamic banking institutions increase their capital level suggesting a prudent behavior on the part of the banks when competition strengthens. In regards to CBs, their risk-taking decisions are dictated by the level of banking concentration whereas increased competition has a negative effect on their risk behavior. This means that regulatory authorities concerned with improving financial stability in the MENA region should proceed differently, depending on the level of concentration and ownership in the banking market. Furthermore, our findings inform regulators and policy makers to set capital requirements at levels that would restrain concentrated banks from taking more risk to increase their profits.

These findings are even more important during the COVID-19 pandemic. On one side, all concerned authorities and regulators should take appropriate measures to sustain the economy by any means rather than accelerating the economic growth. Therefore, regulators responsible for banking sector stability should require a more disciplined approach in bank lending decisions and building a sufficient capital conservation' buffer to limit the impact of downside risk from depletion of capital buffers which can be significant during the pandemic. On the other side, this necessitates a more responsible behavior on behalf of the bank managers when develop their risky strategies.

## Supporting information

**S1 Appendix. List of dependent and explanatory variables.**
(DOCX)

**S2 Appendix. The HH-index and the H-Statistic.**
(DOCX)

**S3 Appendix. Determinants of banking competition (H-Statistic).**
(DOCX)

**S1 Data. Primary data sources used to compute the variables and the sample statistics (Tables 1 and 2).**
(XLSX)

**S1 File. Primary data sources used in the regression analysis run in STATA 11 package (total sample).**
(XLSX)

**S2 File. Primary data sources used in the regression analysis run in STATA 11 package (conventional sample).**
(XLSX)

**S3 File. Primary data sources used in the regression analysis run in STATA 11 package (Islamic sample).**
(XLSX)

## Author Contributions

**Conceptualization:** Miroslav Mateev.

**Data curation:** Muhammad Usman Tariq, Ahmad Sahyouni.

**Formal analysis:** Muhammad Usman Tariq, Ahmad Sahyouni.

**Investigation:** Ahmad Sahyouni.

**Methodology:** Miroslav Mateev, Muhammad Usman Tariq.

**Project administration:** Miroslav Mateev.

**Resources:** Miroslav Mateev.

**Software:** Miroslav Mateev.

**Supervision:** Miroslav Mateev.

**Validation:** Miroslav Mateev, Muhammad Usman Tariq, Ahmad Sahyouni.

**Visualization:** Miroslav Mateev, Muhammad Usman Tariq, Ahmad Sahyouni.

**Writing – original draft:** Miroslav Mateev.

**Writing – review & editing:** Miroslav Mateev, Muhammad Usman Tariq, Ahmad Sahyouni.

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
