## [Decision Letter · Decision Letter 0]

27 Apr 2021

PONE-D-21-09292

Competition, capital growth and risk-taking in emerging markets: Policy implications for banking sector stability during COVID-19 pandemic

PLOS ONE

Dear Dr. Mateev,

Thank you for submitting your manuscript to PLOS ONE. After careful consideration, we feel that it has merit but does not fully meet PLOS ONE’s publication criteria as it currently stands. Therefore, we invite you to submit a revised version of the manuscript that addresses the points raised during the review process.

The manuscript has merit, showing an interesting quantitative analysis, but there are necessary several revisions with reference to study motivation and discussion, as well as English language.

We look forward to receiving your revised manuscript.

Kind regards,

Stefan Cristian Gherghina, PhD. Habil.

Academic Editor

PLOS ONE

Journal Requirements:

2. Please include your tables as part of your main manuscript and remove the individual files. Please note that supplementary tables should be uploaded as separate "supporting information" files.

Reviewers' comments:

Reviewer's Responses to Questions

**Comments to the Author**

1. Is the manuscript technically sound, and do the data support the conclusions?

Reviewer #1: Yes

Reviewer #2: Yes

2. Has the statistical analysis been performed appropriately and rigorously? 

Reviewer #1: Yes

Reviewer #2: Yes

3. Have the authors made all data underlying the findings in their manuscript fully available?

Reviewer #1: Yes

Reviewer #2: No

4. Is the manuscript presented in an intelligible fashion and written in standard English?

Reviewer #1: Yes

Reviewer #2: Yes

5. Review Comments to the Author

Reviewer #1: The research done by Miroslav Mateev and colleagues represents an excellent summary of the competition, capital growth and risk-taking in emerging markets. It is well presented with a thorough inclusion of pertinent studies. It is also presented from an unbiased perspective.

However, there are still a few typographical errors here and there. Furthermore, motivation for using the H-Statistic and the HH-Index to analyze the relation between competition and capital ratios should be provided.

Reviewer #2: When investigating how banking competition and capital level impact on the risk taking behavior of banking institutions, it is important not to ignore the possible effects from relationship lending. Previous literature suggests that relationship lending is a popular lending technology often used by local banks in lending to small businesses (e.g., Petersen and Rajan, 1994; Berger and Udell, 1995). Relationship lending is extremely relevant in time of crisis (Berger, et al. 2020), and when a market has significant information asymmetry (Liberty, Sturgess, and Sutherland, 2017, Wang, 2019). It is crucial to discuss about the role of relationship lending in bank competition in MENA region where conventional banks and Islamic banks coexist. In particular, it would be beneficial to identify banks that are more engaged in relationship lending versus banks that are more specialized in arm's length lending, and test the implications in market with different levels of information asymmetries across 18 countries. I believe authors would be able to make clear contribution by differentiating the effects among Islamic and conventional banks.

References:

Berger, A.N., Bouwman, C.H., Norden, L., Roman, R.A., Udell, G.F. and Wang, T., 2020. Is a Friend in Need a Friend Indeed? How Relationship Borrowers Fare during the COVID-19 Crisis. How Relationship Borrowers Fare during the COVID-19 Crisis, Working paper.

Berger, A. N., & Udell, G. F. 1995. Relationship lending and lines of credit in small firm finance. Journal of business, 351-381.

Petersen, Mitchell A., and Raghuram G. Rajan. 1994, The benefits of lending relationships: Evidence from small business data." The Journal of Finance 49: 3-37.

Liberti, José, Jason Sturgess, and Andrew Sutherland, 2017. Information sharing and lender specialization: Evidence from the US commercial lending market. Working Paper SSRN Journal. http://dx. doi. org/10.2139/ssrn. 2965830.

Wang, T., 2019. To build or to buy? The role of local information in credit market development. Management Science, 65(12), pp.5838-5860.

6. PLOS authors have the option to publish the peer review history of their article (what does this mean?). If published, this will include your full peer review and any attached files.

Reviewer #1: No

Reviewer #2: No

---

## [Author Response · Author response to Decision Letter 0]

31 May 2021

Reviewer(s)' Comments to Author:

Reviewer: 1

Comments:

1. However, there are still a few typographical errors here and there. 

Answer:

We thank the reviewer for this comment. Following his recommendations, we went through a thorough proof-reading process and corrected all gramma and language errors. Additionally, we re-write the introduction to show the rationale and motivation behind our study. We redefine our contribution to the empirical literature and clearly explain how our study differs from previous similar research.

2. Furthermore, motivation for using the H-Statistic and the HH-Index to analyze the relation between competition and capital ratios should be provided.

Answer:

We thank the reviewer for this comment and suggestion. We provide the following explanation to address his comment. 

The Panzar and Rosse (1987)’s H-Statistic is designed to differentiate between competitive, monopolistically competitive, and monopolistic markets. Claessens and Laeven (2004) argue that the H-Statistic is a more appropriate measure for the degree of competition than other proxies for competitive conduct, and Shaffer (2004) notes that the H-Statistic is superior to other measures of competition, because it is derived from profit-maximizing equilibrium conditions. The measure is based on a general banking market model which determines equilibrium output and the number of banks by maximizing profits at both the firm and the industry level. Vesala (1995) shows that the H-Statistic is an increasing function of the demand elasticity, suggesting that as the index increases, less market power is exercised on the part of the banks. This implies that the magnitude of the H-Statistic can serve as a measure for the degree of competition assuming that the bank faces a demand with constant elasticity and a Cobb-Douglas production technology (Schaeck and Čihák, 2007). To obtain the value for the H-Statistic for each year, it is common in the literature to estimate a reduced-form model whereby output is regressed against factor input prices and some control variables that shift the bank’s revenue function (Anzoategui at al., 2010). In this study, to approximate the H-statistic empirically, we use a set up similar to Bikker and Haaf (2002), Claessens and Laeven (2005), and Schaeck et al. (2006). For more information on how the H-Statistic is calculated, please refer to the Supporting information file (S2 Appendix).

The theoretical justification for using concentration as a measure of competition comes from the so called Structure-Conduct-Performance (SCP) paradigm, which postulates that fewer and larger firms (higher concentration) are more likely to engage in anticompetitive behavior (Berger, 1995). The SCP hypothesis supports the notion that high concentrated firms are more competitive and profitable and have more market power in the framework of collusion. The SCP examines the competition conditions by using ratios of concentration of largest firms and Herfindahl-Hirschman index (HHI) that characterize market structure. SCP paradigm is criticized on the assumption that causality is from structure to performance, though it is argued that conduct and performance can affect market structure. Also, the limit of the traditional measures is that the calculation of the degree of competition is done from indirect proxies such as market structure or market shares. 

There is an intensive research on banking competition in the MENA region that uses a variety of structural and non-structural measures (ratios of concentration, HH-index, PR-H statistics and Lerner index) to examine bank competitiveness and market power. For example, Al-Muharrami et al. (2006) use these same indicators to study the structure of banking industry in the Gulf countries and estimate the power of market of these banks during the period 1993-2002. The study finds that a group of countries (Kuwait, Saudi Arabia and United Arab Emirates) have moderately concentrated banking markets, while others (Qatar, Bahrain and Oman) have very concentrated banking markets. Turk Ariss (2010) examines the conditions of competition in 12 countries in the MENA region during the period 2000-2006, and finds that the competition of the banking sector in the region is lower with regard to the other regions and it did not improve during these last years. A study by Hamza and Kachtouli (2014) indicates that both Islamic and conventional markets are characterized by a monopolistic competition and the Islamic banking institutions possess a high degree of market power. Based on the findings of prior research that uses the H-Statistic and the HH-Index as measures of market competitiveness, we analyze the relationship between competition and capital ratios (as measure of financial soundness) of banks in the MENA region using these two specific measures. This allows us to provide some answers related to the role of competitive conditions in the Islamic and conventional markets in the MENA region and the impact of market structure on their financial stability.

Reviewer: 2

Comments:

1. When investigating how banking competition and capital level impact on the risk taking behavior of banking institutions, it is important not to ignore the possible effects from relationship lending. Previous literature suggests that relationship lending is a popular lending technology often used by local banks in lending to small businesses (e.g., Petersen and Rajan, 1994; Berger and Udell, 1995). Relationship lending is extremely relevant in time of crisis (Berger, et al. 2020), and when a market has significant information asymmetry (Liberty, Sturgess, and Sutherland, 2017, Wang, 2019). 

Answer:

We thank the reviewer for this comment and suggestion. We provide the following explanation to address his comment. 

The research on relationship lending focuses on whether relationship borrowers fare better or worse in their loan contract terms (e.g., interest rate spread, collateral, maturity, and loan amount) than other borrowers. In this content, Berger et al. (2020) investigates whether borrowers enjoy the bright side or suffer the dark side of their banking relationships during the COVID-19 crisis. Their results are consistent with the empirical literature that reports dominance of the dark side of relationships during the pandemic crisis; the findings hold across different loan contract terms, relationship measures, COVID-19 shocks, and loan types. The conclusion is that banks do not appear to be “friends indeed with their relationship borrowers in need” (Berger et al., 2020, p.30). The empirical literature generally suggests that small firms are more likely to benefit from relationship lending and small banks are more likely to deliver such benefits. In the early studies in this literature, Petersen and Rajan (1994) find no statistically significant association between the strength of the bank-borrower relationship and business loan pricing, while Berger and Udell (1995) provide evidence to the notion that small firms with longer banking relationship borrow at lower rates and are less likely to pledge collateral than are other small firms. Viewing small banks with high shares of commercial lending to be relationship lenders, DeYoung et al. (2015) find that small relationship lenders cushioned credit crunch problems for small businesses during the Global Financial Crisis (GFC). A recent study of Berger et al. (2020), however, provides limited support for the bright side for smaller firms and smaller banks. 

Credit relationships are usually shaped by the size and structure of the lender (Berger et al. 2005, Berger and Udell 2006; Liberti and Mian 2009). Small lenders are likely to invest in relationships and employ monitoring technologies specific to the given sector, while large lenders are inclined to employ monitoring technologies that are scalable and transferable across markets (Liberti et al. 2017). In the same context, information sharing facilitates entry into new markets and allows larger lenders to have better access to small borrowers. Liberti et al. (2017) argue that if information sharing facilitates entry to new markets, one should observe lenders allocating credits in a manner that suggests they use the credit file information available to them. Therefore, information sharing has implications not only for the geographic and industry composition of the lender’s portfolio, but also for the size of their typical client. In addition to the information sharing effect, the extant research investigates the role of local information in banking and show it is a crucial factor, considering the severe information asymmetries that exist in the intermediation process. Theory shows that a lack of direct access to local information is a disadvantage for banks seeking to enter a new market (Dell’Ariccia et al., 1999). Thus, the question of how banks obtain local information when entering a distant markets, where information shortage is a main concern, becomes of utmost importance. A recent study by Wang (2019) finds that banks entering a new market gain access to local information by building branches and ‘poaching’ incumbent bank employees. Some of these branches serve as a focal point for SME lending. 

2. It is crucial to discuss about the role of relationship lending in bank competition in MENA region where conventional banks and Islamic banks coexist. In particular, it would be beneficial to identify banks that are more engaged in relationship lending versus banks that are more specialized in arm's length lending, and test the implications in market with different levels of information asymmetries across 18 countries. I believe authors would be able to make clear contribution by differentiating the effects among Islamic and conventional banks.

Answer:

We thank the reviewer for this comment and suggestion. We provide the following explanation to address his comment. 

MENA banks also regard the SME segment as potentially profitable, and most banks are already engaged in SME lending to some degree. The drivers that motivate banks to engage in SME lending include the potential profitability of the SME market, the saturation of the large corporate market, the need to enhance returns, and the desire to diversify risks (Rocha et al. 2011). Larger banks have not played a significant role in SME finance in the MENA region, but banks with a larger branch network and these that have set up SME units seem to provide more SME lending, suggesting that relationship lending may still be important in a region where financial infrastructure remains generally deficient. Creditors perceive high risks in SME lending that can only be partially offset through greater reliance on relationship lending, or through the use of other lending techniques such as leasing and factoring, or even through access to a guarantee scheme.

The research on relationship lending in the MENA region is very limited and provides mixed evidence. For example, a study by Rocha et al. (2011) reports that banks in the MENA region still seem to rely on relationship lending, possibly to compensate for the weak financial infrastructure and information asymmetries. The analysis of distribution channels used by banks to service SMEs points to the importance of branches offering services that are tailored to SME needs, which may reflect the continuing importance of ‘relationship banking’. However, it is not clear if the presence of an SME unit by itself means that the bank has moved from relationship lending to transactional lending. Finally, the study concludes that banks use most probably the relationship lending to overcome information asymmetries and the opaqueness of SMEs in the MENA region. Another survey on banks in the MENA region done by the OECD-MENA Women’s Business Forum (WBF) in collaboration with the Union of Arab Banks (UAB) in 2013 (WBF, 2013), reveals that the majority of banks, including banks which reported to have not provided loans to women-owned businesses, possess SME teams (or units) of varying sizes and capacity (the smallest comprising 1 person, the largest with 33 employees). The existence of such teams would suggest a corresponding strategy for SME lending, possibly with customized internal scoring models and specialized products at the relevant banks. Whether this also points to a higher rate of relationship lending is unclear based on existing data from the survey.

One peculiarity of the modern banking systems in the MENA region is the often explicit government support and protection. For example, the governments of most GCC states have significant (often controlling) interests in most of the major banks and established ties with the extended royal families (Gordon, 2018). With few exceptions, most banks in the MENA region, whether in the GCC or not, have historically engaged in conservative lending and maintain a stable base of depositors. These banks have generally focused on large, established clients and the public sector, and they have been reluctant to expand lending in riskier business lines, such as small and medium enterprise or mortgage finance. Therefore, we may view the relationship lending also in the context of the explicit ‘symbiotic’ relationship between the government and its connected families where many of the banks serve the interests of royal families.

Our analysis of the existing empirical literature indicates that there is no unambiguous answer to the question of “to what extend the relationship lending indeed contribute to SME lending”. Typically, the empirical studies examine whether banks are engaged in relationship lending (Berger et al., 1995, Berger and Udell, 1996; Sengupta, 2007), if credit relationships are shaped by the size and structure of the lender (Berger et al. 2005; Berger and Udell 2006; Liberti and Mian 2009), or whether information sharing facilitates entry to new markets (Doblas-Madrid and Minetti, 2013; Sutherland, 2016; Liberti et al., 2017). Accordingly, the studies use different measures (proxies) of “information” in their regression analysis. For example, Liberti et al. (2017) use log number of contracts recorded in the bureau (the U.S. equipment finance credit bureau, PayNet), measured at the collateral type-quarter level. Rocha et al. (2011) is using a variable that controls either for number of branches for a bank or represents a dummy variable which takes a value 1 if bank has a separate unit for SME clients and 0 otherwise. While such data exists for different regions (e.g., GCC vs non-GCC countries), information for Islamic and conventional banks with dedicated SME units is missing, which prevent us from investigating the relationship lending effect across these two banking systems. A request was sent to Bureau Van Dijk’s Orbis Bank Focus which is the main source of data for banks in our sample and received a negative answer. Following Rocha et al. (2011), our future research will consider gathering such information for a number of banks in different countries in the MENA region that we consider to be engaged in relationship lending. This will require the design and implementation of a survey of bank lending to SMEs in the MENA region.

Regarding the implications of relationship lending in markets with different levels of information asymmetries across the MENA countries, the existing literature on asymmetric information is mainly geared toward understanding the main principles guiding Islamic finance such as applying profit and loss sharing and avoiding excessive uncertainty or gharar (Medhioub & Cahffai, 2018). The application of these principles necessitates the fair distribution of information between the parties involved in Islamic financial contracts. In Islamic financial contracts the level of asymmetric information tends to vary from one contract to another. Nagaoka (2010) states that one of the main reasons partnership (musharaka)-based contracts are less attractive is the high asymmetric information attached to these contracts compared to other financial agreements. High banking reserves and capital will provide more assurance to banks against asymmetric information, particularly for those who are placing their funds under less restricted financial contracts like mudaraba. Benamraoui & Alwarda (2019) suggests that Islamic financial contracts are subject to different type of asymmetric information (gharar) -related problems at both the ex-ante and ex-post stages of the lending process. Taking into account the portfolio of Islamic banks and in order to minimize losses caused by the asymmetric information, Islamic banks need to use more secure financing, particularly with SMEs, which tend to be more financially venerable when the economy is in decline. Islamic banks can also adjust their loan pricing to reflect the new lending risks such as credit and market risk and pass some of their costs to borrowers. 

References used:

1. Gordon, M. (2018). The Next Banking Crisis Will Not Be in the Middle East, Middle East Program, Viewpoint No 128, Wilson Center.

2. Rocha, R., Farazi, S., Khouri, R. & Pearce, D. (2011). The status of bank lending to SMEs in the Middle East and North Africa region: the results of a joint survey of the Union of Arab Bank and the World Bank, Policy Research Working Paper Series 5607, Washington DC: The World Bank.

3. Benamraoui, A. & Alwarda, Y. (2019). Asymmetric Information and Islamic Financial Contracts. International Journal of Economics and Finance 11(1), 96-108.

4. Berger, A.N., Bouwman, C.H., Norden, L., Roman, R.A., Udell, G.F. & Wang, T. (2020). Is a Friend in Need a Friend Indeed? How Relationship Borrowers Fare during the COVID-19 Crisis, Working paper. Federal Reserve Bank, Philadelphia (April 9, 2021) http://dx.doi.org/10.2139/ssrn.3755243 .

5. Berger, A.N., Miller, N.H., Petersen, M.A., Rajan, R.G. Stein, J.C. (2005). Does Function Follow Organizational Form? Evidence from The Lending Practices of Large and Small Banks. Journal of Financial Economics 76, 237-269

6. Berger, A. N., & Udell, G. F. (1995). Relationship lending and lines of credit in small firm finance. Journal of business, 68 (3), 351-381.

7. Berger, A., (1995). The profit-structure relationship in banking: Tests of market-power and efficient structure hypotheses. Journal of Money, Credit, and Banking 27(2), 404-431.

8. DeYoung, R., Gron, A., Torna, G., Winton, A. (2015). Risk Overhang and Loan Portfolio Decisions: Small Business Loan Supply before and during the Financial Crisis. Journal of Finance 70, 2451–2488.

9. Dell'Ariccia, G., Friedman, E., & Marquez, R. (1999). Adverse selection as a barrier to entry in the banking industry. The RAND Journal of Economics, 515-534.

10. Doblas-Madrid, A., & Minetti, R. (2013). Sharing information in the credit market: Contractlevel evidence from US firms. Journal of Financial Economics, 109(1), 198-223.

11. Petersen, M. A., & Rajan, R. G. (1994). The benefits of lending relationships: Evidence from small business data. The Journal of Finance 49: 3-37.

12. Liberti, J. & Mian, A. (2009). Estimating the effect of hierarchies on information use. Review of Financial Studies 22, 4057-4090

13. Liberti, J., Sturgess, J., & Sutherland, A. (2017). Information sharing and lender specialization: Evidence from the US commercial lending market. Working Paper, SSRN Journal. http://dx.doi.org/10.2139/ssrn. 2965830.

14. Medhioub, I., & Chaffai, M. (2018). Islamic finance and herding behaviour: An application to Gulf Islamic stock markets. Review of Behavioral Finance 10(2), 192-206. https://doi.org/10.1108/RBF-02-2017-0014

15. Nagaoka, S. (2010). Reconsidering mudarabah contracts in Islamic finance: What is the economic wisdom (hikmah) of partnership-based instruments? Review of Islamic Economics 13(2), 65-79.

16. Shaban, M., Duygun, M., Anwar, M., & Akbar B. (2014). Diversification and banks’ willingness to lend to small businesses: Evidence from Islamic and conventional banks in Indonesia. Journal of Economic Behavior and Organization 103, S39–S55.

17. Sutherland, A. (2016). The Economic Consequences of Borrower Information Sharing: Relationship Dynamics and Investment. Working Paper, DOI:10.2139/SSRN.2626924.

18. Sengupta, R. (2007). Foreign Entry and Bank Competition. Journal of Financial Economics, 84(2), 502-528.

19. Wang, T. (2019). To build or to buy? The role of local information in credit market development. Management Science, 65(12), pp.5838-5860.

20. IMF (2019). Financial Inclusion of Small and Medium-Sized Enterprises in the Middle East and Central Asia, International Monetary Fund, Middle East and Central Asia Department Paper Series No 19/02, Washington DC: International Monetary Fund.

21. WB (2016). What’s Holding Back The Private Sector In Mena? Lessons From The Enterprise Survey. European Bank for Reconstruction and Development, the European Investment Bank, and the International Bank for Reconstruction and Development /The World Bank, Available at: econ_mena_enterprise_survey_en.pdf.

22. OECD-MENA Women’s Business Forum (2013). Exploring bank financing for women entrepreneurs in the MENA region A preliminary analysis of survey data on the financing practices of MENA banks, Draft Report. OECD-MENA WBF and Union of Arab Banks (UAB), Available at https://www.oecd.org/mena/competitiveness/Bank%20Finance.pdf.

---

## [Decision Letter · Decision Letter 1]

14 Jun 2021

Competition, capital growth and risk-taking in emerging markets: Policy implications for banking sector stability during COVID-19 pandemic

PONE-D-21-09292R1

Dear Dr. Mateev,

We’re pleased to inform you that your manuscript has been judged scientifically suitable for publication and will be formally accepted for publication once it meets all outstanding technical requirements.

Kind regards,

Stefan Cristian Gherghina, PhD. Habil.

Academic Editor

PLOS ONE

Additional Editor Comments (optional):

Reviewers' comments:

Reviewer's Responses to Questions

**Comments to the Author**

1. If the authors have adequately addressed your comments raised in a previous round of review and you feel that this manuscript is now acceptable for publication, you may indicate that here to bypass the “Comments to the Author” section, enter your conflict of interest statement in the “Confidential to Editor” section, and submit your "Accept" recommendation.

Reviewer #1: All comments have been addressed

Reviewer #2: All comments have been addressed

2. Is the manuscript technically sound, and do the data support the conclusions?

Reviewer #1: Yes

Reviewer #2: Yes

3. Has the statistical analysis been performed appropriately and rigorously? 

Reviewer #1: Yes

Reviewer #2: Yes

4. Have the authors made all data underlying the findings in their manuscript fully available?

Reviewer #1: No

Reviewer #2: Yes

5. Is the manuscript presented in an intelligible fashion and written in standard English?

Reviewer #1: Yes

Reviewer #2: Yes

6. Review Comments to the Author

Reviewer #1: The research done by Miroslav Mateev and colleagues represents an excellent summary of the competition, capital growth, and risk-taking in emerging markets. It is well-grounded with the appropriate literature after revision. It is also presented from an unbiased perspective.

Reviewer #2: (No Response)

7. PLOS authors have the option to publish the peer review history of their article (what does this mean?). If published, this will include your full peer review and any attached files.

Reviewer #1: No

Reviewer #2: No

---

## [Editor Report · Acceptance letter]

16 Jun 2021

PONE-D-21-09292R1 

Competition, capital growth and risk-taking in emerging markets: Policy implications for banking sector stability during COVID-19 pandemic 

Dear Dr. Mateev:

I'm pleased to inform you that your manuscript has been deemed suitable for publication in PLOS ONE. Congratulations! Your manuscript is now with our production department. 

Kind regards, 

on behalf of

Dr. Stefan Cristian Gherghina 

Academic Editor

PLOS ONE